**Summertime fine particulate nitrate pollution in the North China Plain: Increasing trends, formation mechanisms, and implications for control policy**

Liang Wen[1], Likun Xue[1*], Xinfeng Wang[1], Caihong Xu[1], Tianshu Chen[1], Lingxiao Yang[1], Tao Wang[2], Qingzhu Zhang[1], Wenxing Wang[1]

[1] Environment Research Institute, Shandong University, Ji'nan, Shandong, China

[2] Department of Civil and Environmental Engineering, Hong Kong Polytechnic University, Hong Kong, China

*Corresponding to: L. K. Xue, xuelikun@sdu.edu.cn

## Abstract

Nitrate aerosol composes a significant fraction of fine particles and plays a key role in regional air quality and climate. The North China Plain (NCP) is one of the most industrialized and polluted regions in China. To obtain a holistic understanding of the nitrate pollution and its formation mechanisms over the NCP region, intensive field observations were conducted at three sites during summertime in 2014-2015. The measurement sites include an urban site in downtown Ji'nan – the capital city of Shandong Province, a rural site downwind of Ji'nan city, and a remote mountain site at Mt. Tai (1534 m a.s.l.). Elevated nitrate concentrations were observed at all three sites despite distinct temporal and spatial variations. Using historical observations, the nitrate/$PM_{2.5}$ and nitrate/sulfate ratios have statistically significantly increased in Ji'nan (2005-2015) and at Mt. Tai (from 2007 to 2014), indicating the worsening situation of regional nitrate pollution. A multi-phase chemical box model (RACM/CAPRAM) was deployed and constrained by observations to elucidate the nitrate formation mechanisms. The principal formation route is the partitioning of gaseous $HNO_3$ to aerosol phase during the day, whilst the nocturnal nitrate formation is dominated by the heterogeneous hydrolysis of $N_2O_5$. The daytime nitrate production in the NCP region is mainly limited by the availability of $NO_2$, and to a lesser extent by $O_3$ and $NH_3$. In comparison, the nighttime formation is controlled by both $NO_2$ and $O_3$. The presence of $NH_3$ contributes to the formation of nitrate aerosol during the day, while slightly decreasing nitrate

formation at night. Our analyses suggest that controlling $NO_X$ and $O_3$ is an efficient way at the moment to mitigate nitrate pollution in the NCP region, where $NH_3$ is usually in excess in summer. This study provides observational evidence of a rising trend of nitrate aerosol as well as scientific support for formulating effective control strategies for regional haze in China.

## 1.  Introduction

Atmospheric particles are vital players in tropospheric chemistry, regional air pollution, and climate change. High concentrations of fine particles (i.e., $PM_{2.5}$) can reduce visibility (Xu and Penner, 2012), deteriorate air quality (Huang et al., 2014), and are harmful to human health (Xie et al., 2016). They play an essential role in Earth's radiation balance and hence affect climate change, directly by scattering and absorbing the incoming solar radiation (IPCC, 2013) and indirectly by modifying the cloud properties (Ding et al., 2013; Fukushima et al., 2016). Aerosol particles can also serve as a medium for reactive gases to undergo heterogeneous and aqueous phase reactions (Chang et al., 2011). Understanding the chemical composition and sources of atmospheric particles is crucial for quantifying their environmental and health consequences and formulating science-based mitigation strategies.

Particulate nitrate ($NO_3^-$) is a principal chemical component of atmospheric fine particles. It is an oxidation product of nitrogen oxides ($NO_X=NO+NO_2$) in the ambient atmosphere. During the day, the oxidation of $NO_2$ by the hydroxyl radical (OH) produces gaseous nitric acid ($HNO_3$), which then reacts with ammonia ($NH_3$) or other alkaline compounds to form nitrate aerosol (Calvert and Stockwell, 1983). The partitioning of $HNO_3$ between gas and aerosol phases is dependent on ambient temperature, humidity and the abundances of alkaline species (Song and Carmichael, 2001; Wang et al., 2009a; Yao and Zhang, 2012). In dark conditions, the reaction of $NO_2$ and $O_3$ produces the nitrate radical ($NO_3$), which forms an equilibrium with $N_2O_5$ that can be subsequently taken up onto particles to enhance nitrate aerosol (Pathak et al., 2009 and 2011; Brown and Stutz, 2012). The contribution from this pathway is minimized during the day by the rapid photolysis of $NO_3$ and thermal decomposition of $N_2O_5$. The $HNO_3$ partitioning and $N_2O_5$ hydrolysis reactions have been recognized as the major sink pathways of NOx in the troposphere (Dentener and Crutzen,

1993; Liu et al., 2013). There are some other formation routes of nitrate aerosol, such as the uptake of $NO_3$ radicals onto particles and its subsequent aqueous reactions with some water-soluble species (Hallquist et al., 1999; see also Table S1). The ambient formation of nitrate aerosol highly depends on the chemical mix of $NO_X$, $O_3$ and $NH_3$. To date the detailed relationship between nitrate formation and the chemical mix of $NO_x$, $O_3$ and $NH_3$ is still poorly understood. The contribution of $N_2O_5$ hydrolysis pathway tends to show a seasonal dependence with the largest influence in the winter season (Brown and Stutz, 2012; Baasandorj et al., 2017), but will be dependent on the rate of $NO_3$ formation and reaction, the $N_2O_5$ uptake coefficient ($\gamma(N_2O_5)$) and formation yield of $ClNO_2$. Field measurement studies have shown that the $\gamma(N_2O_5)$ is highly variable and disagrees with the laboratory-derived parameterizations (Brown and Stutz, 2012; McDuffie et al., 2018 and Tham et al., 2018). Vertical mixing of air aloft in the residual layer may also contribute to the surface nitrate pollution (Watson et al., 2002; Brown et al., 2006; Pusede et al., 2016; Baasanforj et al., 2017; Prabhakar et al., 2017). Consequently, there are still some remaining questions for better understanding the nitrate formation mechanisms.

China has been suffering from severe haze pollution as a result of its fast urbanization and industrialization processes in the past decades. The North China Plain (NCP), covering the Beijing-Tianjin-Hebei area and surrounding Shandong and Henan provinces, is the most polluted region with the highest annual concentrations of $PM_{2.5}$ in China (http://www.cnemc.cn/kqzlzkbgyb2092938.jhtml). Previous air pollution control in China primarily focused on the reduction of anthropogenic emissions of sulfur dioxide ($SO_2$), given the dominant contributions of sulfate ($SO_4^{2-}$) to the $PM_{2.5}$ and acid deposition (Hao et al., 2000 and 2007). In the last decade, about a 75% reduction of $SO_2$ emissions in China has successfully resulted in decreases in the ambient levels of both $SO_2$ and aerosol $SO_4^{2-}$ in fast-developing regions including the NCP (Wang et al., 2013; Li et al., 2017). In comparison, several recent observational studies have indicated an increasingly important role of aerosol nitrate, which may even dominate summertime haze formation in the NCP region (Wen et al., 2015; Li et al., 2018). A recent modeling study has predicted a significant increase of aerosol nitrate along with the decrease of sulfate during 2006-2015 over eastern China (Wang et al.,

2013). To the best of our knowledge, there are no previous observational reports of increasing nitrate aerosol over northern China. Long-term measurements are necessary to confirm and quantify this trend, and better understand the nitrate formation mechanisms in China.

To achieve a better understanding of the summertime nitrate pollution and its formation mechanism in the NCP region, four phases of intensive observations were conducted at three different sites covering urban, rural and remote areas in 2014 and 2015. The spatial distribution and temporal variation of nitrate aerosol pollution were examined. The data were combined with previous measurements to derive the trends of the "nitrate/$PM_{2.5}$" and "nitrate/sulfate" ratios, confirming the statistically significant increase of regional nitrate pollution during 2005-2015. A multi-phase chemical box model, constrained by in-situ observations, was then deployed to unravel the formation mechanisms of fine particulate nitrate. The impacts of $NO_2$, $O_3$ and $NH_3$ on the regional nitrate formation were finally quantified. Overall, the present study provides the first piece of the observational evidence for the increasing trend of nitrate aerosol in northern China, and our findings have important implications for the future control of regional haze pollution in the NCP region.

## 2. Materials and methods

### 2.1. Study sites

To better understand the regional-scale nitrate pollution and formation processes, four phases of intensive field campaigns were conducted at three sites in the central part of the North China Plain in the summers of 2014-2015. Considering that southerly/southeasterly winds generally prevail in summertime, the three study sites were carefully selected to lie on a southeast-northwest transect (see Figure 1), and to represent typical urban, rural and remote atmospheres of the region. A summary of the measurement locations and periods are shown in Table S2.

The urban site (36.67 °N, 117.06 °E, ~50 m above sea level (a.s.l.)) was located in the downtown area of Ji'nan, the capital city of Shandong province, accommodating more than 7 million inhabitants, ~1.7 million automobiles and many factories. Ji'nan is one of the largest cities in the central NCP, and has been frequently ranked among the worst ten key cities of

China in terms of air quality (http://www.cnemc.cn/kqzlzkbgyb2092938.jhtml). The site is built on the rooftop of a six-floor building in the Central Campus of Shandong University, which is situated in the residential and commercial areas. Details of this site have been provided in our previous publications (Gao et al., 2011; Wang et al., 2015). Two intensive campaigns took place during 5$^{th}$-17$^{th}$ May 2014 and 23$^{rd}$ August - 21$^{st}$ September 2015, respectively. In addition, measurements of aerosol ionic components have been made previously at this site in selected years since 2005 (Yang et al., 2007 and 2012; Gao et al., 2011; Zhu et al., 2015).

The rural site (36.87 ˚N, 116.57 ˚E, ~23 m a.s.l.) was set up at the Chinese Academy of Sciences Comprehensive Station in Yucheng. Although Yucheng belongs to Dezhou city, it serves as a satellite town of Ji'nan. The measurement site is located about 50 km northwest (normally downwind in summer) of downtown Ji'nan (Fig. 1), and thus can be regarded as a receptor site of urban pollution. The instruments were housed in a temperature controlled container that was placed in an open cropland with few anthropogenic emissions nearby (Wen et al., 2015; Zhu et al., 2016). A six-week campaign was carried out here from 2$^{nd}$ June to 16$^{th}$ July 2014.

The remote site (36.26 ˚N, 117.11 ˚E, 1465 m a.s.l.) was installed at the summit of Mt. Tai. Mt. Tai is the highest mountain over the North China Plain (with a peak of 1534 m a.s.l.), and has been widely deployed as the sampling platform to investigate regional air pollution (Gao et al., 2005; Sun et al., 2016). The station was set up in a hotel to the north of the mountain peak with a little lower elevation. It is located approximately 15 km north of Tai'an city (with a population of 5.6 million) and 40 km south of urban Ji'nan (Fig. 1). Detailed descriptions of this site can be found elsewhere (Guo et al., 2012; Shen et al., 2012). In the present study, the measurements were conducted from 23$^{rd}$ July to 27$^{th}$ August 2014. Previous data collected at Mt. Tai in summer 2007 are also analyzed to examine the long-term change in the regional nitrate pollution (Zhou et al., 2010).

**2.2. Measurement techniques**

A Monitor for AeRosols and GAses (*MARGA, ADI20801, Applikon-ECN, Netherlands*) was deployed in the present study to measure continuously, at a time resolution of 1-hour,

inorganic water-soluble ions (i.e., $NO_3^-$, $SO_4^{2+}$, $NH_4^+$, $Cl^-$, etc.) in $PM_{2.5}$ together with acid and alkaline gases (i.e., $HNO_3$, $HCl$, $NH_3$, etc.). The target gases and ions are collected and dissolved by a Wet Rotating Denuder (WRD) and a Steam Jet Aerosol Collector (SJAC), respectively (Brink et al., 2009). The dissolved components are then analyzed by a cationic and an anionic ion chromatography with eluent solutions of methane sulfonic acid (308 mg $L^{-1}$) and $NaHCO_3$ (672 mg $L^{-1}$)-$Na_2CO_3$ (742 mg $L^{-1}$). An internal standard solution of LiBr (4 mg $L^{-1}$) was added automatically into the collected sample solutions to calibrate the detection in each analytic process. Multi-point calibration was performed before and after the field campaigns to examine the sensitivity of the detectors. The detection limits were evaluated as 0.05, 0.04 and 0.05 μg $m^{-3}$ for particulate $NO_3^-$, $SO_4^{2-}$ and $NH_4^+$, and 0.01, 0.01 and 0.07 ppbv for gaseous $HNO_3$, $SO_2$ and $NH_3$, respectively. The MARGA instrument has been deployed in many field studies in the high aerosol loading environment in China (e.g., Wen et al., 2015; Xie et al., 2015).

To achieve a detailed analysis of nitrate formation processes, a large suite of ancillary measurements were concurrently made during the field studies. $PM_{2.5}$ mass concentrations were quantified in-situ by a SHARP monitor (*Model 5030, Thermo Scientific, USA*); particle size and counts in the range of 5-10000 nm were monitored by a wide-range particle spectrometer (*WPS; Model 1000XP, MSP Corporation, USA*); NO and $NO_2$ by a chemiluminescence instrument equipped with an internal molybdenum oxide (MoO) catalytic converter (*Model 42C, Thermo Electron Corporation, USA*); $O_3$ by an ultraviolet absorption analyzer (*Model 49C, Thermo Electron Corporation, USA*); CO by a non-dispersive infrared analyzer (*Model 300EU, API, USA*); $SO_2$ by an ultraviolet fluorescence analyzer (*Model 43C, Thermo Electron Corporation, USA*); meteorological parameters including temperature, relative humidity (RH) and wind sectors by commercial automatic weather stations. All of these techniques have been widely applied in many previous studies, to which detailed information can be referred (Gao et al., 2011; Wang et al., 2012; Xue et al., 2014).

### 2.3. Multi-phase chemical box model

A zero-dimensional chemical box model was configured to simulate the in-situ formation of fine nitrate aerosol. It couples the regional atmospheric chemistry mechanism version 2

(RACM2; including 363 chemical reactions) and the chemical aqueous phase radical mechanism version 2.4 (CAPRAM 2.4; including 438 chemical reactions) to account for gas- and aqueous-phase atmospheric chemistry (Goliff et al., 2013; Herrmann et al., 2000 and 2005). The gas-aqueous phase transfer processes were adopted from the resistance scheme of Schwartz (1986). This model explicitly describes the gas-to-aqueous phase partitioning of various chemical species, which connects the detailed chemical reactions in both gas and aqueous phases. The chemical reactions representing the nitrate formation in the model are outlined in Table S1. Briefly, these reactions can be categorized into three major formation pathways, namely, partitioning of gaseous $HNO_3$ to the aerosol phase, hydrolysis reactions of $N_2O_5$, and aqueous phase reactions of $NO_3$ radicals. The $HNO_3$ partitioning is largely affected by the availability of $NH_3$, since the partitioning of $NH_3$ would decrease the aerosol acidity and hence enhance the partitioning of $HNO_3$ to the aerosol phase (see Table S1). For the $N_2O_5$ hydrolysis process, the uptake coefficient of $N_2O_5$ on particles ($\gamma N_2O_5$) is the parameter with large uncertainty in modeling studies. Recent studies have shown that $\gamma N_2O_5$ tends to be largely variable and significant discrepancy exists between field-derived laboratory-derived parameterizations (Chang et al., 2011; McDuffie et al., 2018 and Tham et al., 2018). The RACM/CAPRAM model doesn't take $\gamma N_2O_5$ into account, but describes explicitly the $N_2O_5$ gas-to-aqueous phase partitioning as well as its subsequent aqueous phase reactions. See Table S1 for the detailed treatment of the $N_2O_5$ hydrolysis processes in the model. We estimated the $\gamma(N_2O_5)$ from the reaction rate for the $N_2O_5$ gas-to-particle partitioning and the measured aerosol surface area concentrations, and derived an average $\gamma(N_2O_5)$ ($\pm$SD) of 0.018$\pm$0.00006 for our selected cases. Such levels are well within the reported range of $\gamma(N_2O_5)$ derived from the field observations in other locations worldwide (e.g., 0.001-0.1), including several polluted areas in northern China (Tham et al., 2018; and references therein). This model has been utilized previously to simulate the nighttime nitrate formation in Beijing and Shanghai (Pathak et al., 2011).

The model calculation requires a large number of variables and parameters, including: 1) gas phase concentrations of NO, $NO_2$, $O_3$, $SO_2$, HCl, $HNO_2$, $HNO_3$, $NH_3$, CO and VOCs, etc.; 2) particulate (or aqueous) phase concentrations of $NO_3^-$, $SO_4^{2-}$, $Cl^-$, $HSO_4^-$, $NH_4^+$, $H^+$, $OH^-$,

etc.; 3) other auxiliary parameters such as temperature, RH, pressure, boundary layer height, particle radius, and aerosol water content, etc. Most of the above parameters were observed in-situ during our intensive measurement campaigns, and the available data were directly used to constrain the model. The measured aerosol ions data such as nitrate, sulfate and ammonium were only used as initial conditions of the model simulation. The model was initialized with the measured nitrate concentration at the beginning of the episodes, and then simulated its formation with constraints of other relevant species. The particle radius was calculated from the measured aerosol number and size distribution with an assumption that all particles were spherical. A hygroscopic growth factor obtained from the NCP region by Achtert et al. (2009) was adopted to take into account the effect of hygroscopic growth on the particle size and surface. Aerosol $H^+$, $OH^-$, $HSO_4^-$, and water content were simulated by a thermodynamic model (E-AIM; http://www.aim.env.uea.ac.uk/aim/aim.php) based on the measured aerosol chemistry data (Clegg et al., 1998; Zhang et al., 2000). The VOC measurements were not made during the present study, and we used the campaign average data previously collected in the same areas during summertime for approximation (Wang et al., 2015; Zhu et al., 2016 and 2017). The detailed VOC species and their concentrations as model input are documented in Table S3. We conducted sensitivity tests with 0.5 or 1.5 times of the initial VOC concentrations, and found that the model simulation was somewhat insensitive to the initial VOC data (the differences in the model-simulated nitrate formation between sensitivity tests and base run were within 12%; see Figure S1). This should be mainly due to the low levels of biogenic VOCs in the study area. Given the lack of in-situ VOC measurements, however, the treatment of VOC data presents a major uncertainty in the present modeling analyses. The boundary layer height, which affects the dry deposition of various constituents, was estimated by the Nozaki method (Nozaki, 1973). The dry deposition velocity of $HNO_3$ was set as 2 cm s$^{-1}$ in the model. With such configuration, dry deposition only presents a minor fraction of the daytime $HNO_3$ sink (<1%), compared to the $HNO_3$ gas-to-particle partitioning.

Simulations were conducted for selected nighttime or daytime nitrate formation cases. The starting time and simulation periods depended on the individual cases. The output data

included particulate nitrate concentrations and reaction rates of the major aerosol formation pathways. In addition, a number of sensitivity simulations were performed to examine the relationships between nitrate formation and its precursors (see Sections 3.3 and 3.4).

## 3.    Results and discussions

### 3.1.    Temporal and spatial variations

Table 1 summarizes the statistics of the aerosol chemical properties, trace gases and meteorological parameters measured at three study sites. It clearly shows the spatial distribution of regional aerosol pollution though elevated levels of $PM_{2.5}$ and major ions were observed at all three sites. The highest $PM_{2.5}$ levels were recorded at the receptor rural site (Yucheng; with campaign average $\pm$ SD of $97.9\pm53.0$ µg m$^{-3}$), followed by the urban (Ji'nan; $68.4\pm41.7$ and $59.3\pm31.8$ µg m$^{-3}$ in 2014 and 2015, respectively) and mountain sites (Mt. Tai; $50.2\pm31.7$ µg m$^{-3}$). Nitrate shows a similar gradient with average concentrations ranging from $6.0\pm4.6$ µg m$^{-3}$ at Mt. Tai to $13.6\pm10.3$ µg m$^{-3}$ at Yucheng. In comparison, $SO_4^{2-}$ shows a slightly different pattern with the lowest levels found in Ji'nan ($12.2\pm7.5$ and $12.7\pm7.9$ µg m$^{-3}$), then Mt. Tai ($14.7\pm8.9$ µg m$^{-3}$) and Yucheng ($23.6\pm13.4$ µg m$^{-3}$). Chloride showed comparable levels in urban Ji'nan ($1.3\pm2.1$ and $1.3\pm1.7$ µg m$^{-3}$) and rural Yucheng ($1.2\pm1.2$ µg m$^{-3}$), with a relatively lower level at Mt. Tai ($0.7\pm0.5$ µg m$^{-3}$). For $NO_2$, an anthropogenic emission indicator and a major precursor of $NO_3^-$, the highest mixing ratios were determined in urban Ji'nan, followed by Yucheng and Mt. Tai. The nitrate oxidation ratio (NOR), defined as the molar ratio of $NO_3^-$ to $NO_3^-+NO_X$, shows an opposite pattern with the lowest values in Ji'nan ($0.11\pm0.07$ and $0.16\pm0.08$ in 2014 and 2015) and highest levels at Mt. Tai ($0.39\pm0.20$). This indicates the different extent of chemical processing of air masses in different types of areas. The air masses sampled at Mt. Tai were more aged and longer air transport allowed more time for chemical processing. The above regional gradients of air pollution are mainly owing to the spatial distribution of anthropogenic emissions and different chemical aging of air masses. It should be noted that these measurements were not conducted simultaneously, and thus differences in the reported data at three study sites can be expected in view of the potential differences in the meteorological conditions which affect atmospheric mixing and transport processes. However, the spatial distributions of emissions, atmospheric chemical

and physical processes are still believed to be the major factor shaping the observed regional pattern of aerosol pollution.

Table 1 also illustrates some homogeneity of the regional aerosol pollution and chemistry in the NCP region. First, secondary inorganic ions (i.e., $SO_4^{2-}$, $NO_3^-$ and $NH_4^+$) accounted for on average 41%-56% of $PM_{2.5}$ at three sites, indicating their dominant roles in the aerosol composition and regional haze. Second, $NO_3^-$ alone presented an important fraction of fine particles, and the $NO_3^-/PM_{2.5}$ ratios were nearly uniform over the region, with average values of 11%-14% at all three sites. Based on field measurements in January 2013, Huang (2014) also reported that $NO_3^-$ accounted for 12%, 14% and 13% of $PM_{2.5}$ in Beijing, Shanghai and Guangzhou, with a smaller ratio (7%) recorded in a western city (Xi'an). At the surface sites (i.e., Ji'nan and Yucheng) in the present study, the molar concentrations of $NO_3^-$ were even comparable to $SO_4^{2-}$, with mean $NO_3^-/SO_4^{2-}$ ratios of 0.93-1.04. In comparison, the $NO_3^-/SO_4^{2-}$ ratio was lower (0.62±0.33) at Mt. Tai, which should be due to the longer lifetime of sulfate aerosol and frequent transport of power plant plumes to the mountain site (Wang et al., 2017). Finally, $NH_4^+$ was generally in excess in $PM_{2.5}$. The average excess $NH_4^+$ (excess $NH_4^+=18*([NH_4^+]-1.5*[SO_4^{2-}]-[NO_3^-]-[Cl^-])$) was calculated in the range of 0.9-4.3 μg m$^{-3}$ at our three study sites. This highlights the $NH_3$-rich chemical environment of the NCP region, and the abundant $NH_3$ may significantly affect the formation of nitrate aerosol (see Section 3.4).

Figure 2 clearly shows the distinct diurnal variation patterns of $NO_3^-$ and $NO_2$ at the three different types of sites. In urban Ji'nan, $NO_3^-$ showed a maximum level in the early morning (8:00 local time; LT) with a secondary peak in the afternoon (15:00 LT). At Yucheng, the average diurnal profile displays a continuous nitrate formation process throughout the nighttime with a $NO_3^-$ increase of 16.9 μg m$^{-3}$ from 16:00 to 8:00 LT, followed by a sharp decrease during the day with a trough in the late afternoon (16:00 LT). The absolute nighttime $NO_3^-$ levels were higher than the daytime concentrations at Ji'nan and Yucheng, owing to the dilution within the developed planetary boundary layer (PBL) and thermal decomposition of aerosol in high temperature conditions during the day. An opposite diurnal profile was observed at Mt. Tai, which showed an $NO_3^-$ increase throughout the daytime and high

concentrations remaining in the early evening. The daytime increase was due to the development of the PBL and the mountain-valley breeze, both of which could carry the boundary layer pollution aloft, and the elevated evening levels should be ascribed to the regional transport of polluted plumes to the mountain top (Sun et al., 2016). Overall, $NO_2$ showed similar diurnal variations with $NO_3^-$, and the $NO_3^-$ concentration peaks generally lagged behind $NO_2$, suggesting the role of $NO_2$ in the nitrate formation as a precursor. Inspection of diurnal variations day by day also revealed frequent nitrate formation at all three sites during nighttime and also during the day (especially at Mt. Tai). We then selected a dozen of nitrate formation cases for detailed modeling analyses in Section 3.3.

## 3.2.  Trend over 2005-2015

Figure 3 shows the increasing trends of $NO_3^-$ in $PM_{2.5}$ in the past decade over the NCP region. Intensive measurements of aerosol ionic species have been made by our group in urban Ji'nan in selected years since 2005 (Yang et al., 2007 and 2012; Gao et al., 2011; Zhu et al., 2015) and at Mt. Tai in 2007 (Zhou et al., 2010), and these previous data were combined with the more recent observations in the present study to derive the decadal trends. To eliminate the interference of inter-annual variation of weather condition on the absolute concentrations, we focused on the ratios of $NO_3^-/PM_{2.5}$ and $NO_3^-/SO_4^{2-}$ for the trend analysis. In urban Ji'nan, the fraction of $NO_3^-$ in $PM_{2.5}$ has increased at a rate of 0.9% per year over 2005-2015 ($p<0.01$). A similar increasing rate (0.7% per year) was also derived at Mt. Tai from data collected in 2007 to 2014, affirming the statistically significant increase of fine particulate nitrate over the region. At the same time, the $SO_4^{2-}$ in $PM_{2.5}$ has statistically significantly declined in urban Ji'nan (-0.7% per year) and at Mt. Tai (-1.3% per year; figures not shown), as a result of the strict control of $SO_2$ emissions in China. As a result, the molar ratio of $NO_3^-/SO_4^{2-}$ has increased at a rate of 0.09 per year in Ji'nan during 2005-2015 ($p<0.01$) and 0.05 per year at Mt. Tai from 2007 to 2014 (Fig. 3b).

We also examined the trends in the absolute concentrations of $PM_{2.5}$, nitrate and sulfate in urban Ji'nan and at Mt. Tai (see Fig. S2). As expected, the ambient concentrations of $PM_{2.5}$ (-6.3 and -1.4 $\mu g\ m^{-3}\ yr^{-1}$) and $SO_4^{2-}$ (-2.1 and -1.2 $\mu g\ m^{-3}\ yr^{-1}$) have rapidly decreased at both locations during the past decade, which should be largely attributed to the stringent control of

$SO_2$ emissions and primary particles. In comparison, the absolute concentrations of $NO_3^-$ showed an increasing trend with average rates of change of 0.39 and 0.29 $\mu g\ m^{-3}\ yr^{-1}$ at Ji'nan and Mt. Tai, respectively. This confirms the increase of absolute nitrate aerosol pollution in the NCP region. Nevertheless, the available observations since 2011 also showed a decrease in the absolute levels of nitrate aerosol in Ji'nan. This trend may be true considering the strict $NO_x$ emission control of China since 2011, but it may be also partly interfered by the higher aerosol pollution observed during the campaign of 2011 with unfavorable meteorological conditions. More measurement efforts are urgently needed to further examine the recent trend of nitrate aerosol after 2011 and evaluate the impact of the $NO_x$ emission control of China.

Our observations provide the direct evidence of a statistically significant increase of summertime nitrate aerosol in the NCP region along with a decrease of sulfate in the last decade. The comparable contributions of $NO_3^-$ and $SO_4^{2-}$ to $PM_{2.5}$ suggest the gradual shift of the secondary inorganic aerosol type from $SO_4^{2-}$-dominant to $NO_3^-$-and-$SO_4^{2-}$-dominant. A recent modeling study also predicted an increase of nitrate with a decrease of sulfate from 2006 to 2015 over the entire eastern China (Wang et al., 2013). A more recent observational study at two sites (Beijing and Xinxiang) in the NCP region indicated the important contributions of nitrate in $PM_1$ and its driving role in the summertime haze pollution (Li et al., 2018). Overall, nitrate has been playing an increasingly important role in the haze pollution in northern China. In recent years, the strict anti-pollution measures implemented by the central government have led to a significant reduction in the primary $PM_{2.5}$ in the NCP, while secondary aerosols such as nitrate are still at high levels and present the major challenge for further mitigation of haze pollution (http://www.cnemc.cn/kqzlzkbgyb2092938.jhml). Nitrate and its precursors should be the next major target for the future control of regional haze pollution in China.

**3.3. Nitrate formation mechanisms**

Multi-phase chemical modeling was then conducted for typical nitrate formation events to understand the formation mechanisms of fine particulate nitrate at three study sites. The selected cases met the following criteria: 1) the nitrate formation (accumulation) process should last for a considerable time period (i.e., at least three hours); 2) the observed NOR

(NOR=[NO$_3^-$]/([NO$_3^-$]+[NO$_x$])) was increasing throughout the event; 3) the meteorological conditions were stable with constant wind direction or a calm condition and without wet deposition; 4) the data in the early morning period (i.e., 06:00-09:00 LT) were excluded from analyses to roughly eliminate the potential influence from downward mixing of air aloft to the surface sites. A total of 21 nitrate formation events were finally sorted out, including 10 daytime cases (3, 3 and 4 in Ji'nan, Yucheng and Mt. Tai) and 11 nighttime ones (3, 5 and 3 in Ji'nan, Yucheng and Mt. Tai). Details of these selected cases are provided in the supplement (see Table S4).

Figure 4 compares the model-simulated versus observed nitrate enhancements for the daytime cases, and also presents the contributions of the major nitrate formation pathways. Generally, the model reproduced well the observed nitrate formation, with a strong positive correlation between simulations and observations (with a reduced major axis (RMA) slope of 0.90 and r$^2$ of 0.60; see Fig. S3). The partitioning of HNO$_3$ gas to the particulate phase was clearly the predominant daytime formation pathway of nitrate aerosol, with average contributions of 96%, 95% and 94% at the urban, rural and remote sites, respectively. Hydrolysis of N$_2$O$_5$ contributed to the remaining (4-6%), and the direct uptake and aqueous-phase reactions of NO$_3$ radicals was negligible.

The modeling results for the nighttime cases are shown in Figure 5. The model also worked reasonably well for the simulation of nitrate formation at night, as indicated by the strong positive correlation between the simulated and observed NO$_3^-$ enhancements with a RMA slope (simulation/observation) of 1.60 and r$^2$ of 0.93 (see Fig. S3). Figure S4 shows that the model reproduced the absolute concentrations of nitrate for two specific typical cases. The hydrolysis reaction of N$_2$O$_5$ turned out to be the overwhelming formation pathway at nighttime, with mean contributions of 94%, 98% and 91% at the urban, rural and remote sites, respectively. Other processes such as the HNO$_3$ partitioning and aqueous reactions of NO$_3$ radicals were minor routes. These results are in line with the previous studies that have assessed the nitrate formation pathways. For example, Pathak et al. (2011) found that the N$_2$O$_5$ hydrolysis contributed to 50%-100% of the nocturnal nitrate formation in Beijing and Shanghai. Based on the field measurements of N$_2$O$_5$ and related species, Wang et al. (2017)

suggested that the $N_2O_5$ hydrolysis contributed comparably to or even higher than the partitioning of $HNO_3$ to nitrate formation in Beijing at a daily basis. Overall, the significant roles of $HNO_3$ partitioning and $N_2O_5$ hydrolysis in nitrate formation have been well outlined (Brown and Stutz, 2012).

The budgets of nitrate formation were almost the same among the three study sites. This indicates the regional homogeneity of formation mechanism of fine nitrate aerosol over the NCP region. The formation of $HNO_3$ and its subsequent partitioning to the aerosol phase is the principal formation route during the day, while the hydrolysis reactions of $N_2O_5$ on the particles play a dominant role during the night. This is in line with the current understanding

that the oxidation of $NO_2$ by OH forming $HNO_3$ and heterogeneous reactions of $N_2O_5$ present the major $NO_X$ sinks during the daytime and nighttime, respectively (Liu et al., 2013).

  According to the above identified major formation pathways, the nitrate formation can be influenced by the availability of $NO_x$, $O_3$ and $NH_3$. $NO_x$ are direct precursors of nitrate formation. $O_3$ is a major oxidant and supplier of OH radicals during the day, and is also a

precursor of $N_2O_5$ at night. $NH_3$ may prompt the partitioning of $HNO_3$ to the aerosols, and alter the aerosol acidity that affects not only the partitioning of $HNO_3$ but also the hydrolysis of $N_2O_5$. Therefore, we further examined the dependence of nitrate formation to $NO_2$, $O_3$ and $NH_3$ at the three sites by sensitivity analyses. Sensitivity modeling calculations were conducted by adjusting the concentrations of the target species ($NO_2$ or $O_3$ or $NH_3$) by $X$

20  times (i.e., 0, 0.1, 0.2, 0.3, 0.4, 0.5, 0.8, 1.2 and 1.5), and the other settings remained unchanged with the base simulations. The difference in the simulated $NO_3^-$ concentrations between base and sensitivity runs should reflect the impact of the change in the target species on nitrate formation. The sensitivity modeling results for the daytime cases are documented in Figure 6. Similar results were derived from the three different study areas. During the day,

the nitrate formation was the most sensitive to $NO_2$, a necessary precursor of $NO_3^-$ aerosol. It was also sensitive to a lesser extent to $O_3$, which is a major OH source and thus affects the gaseous $HNO_3$ formation. An interesting finding was the dependence of nitrate formation to the abundance of $NH_3$. Adjusting (neither increasing nor decreasing) the currently measured $NH_3$ concentrations by up to 50% would not lead to significant changes in the model

simulated $NO_3^-$, whilst further reduction of $NH_3$ (c.a., more than 50%-80%) would result in a significant decrease of $NO_3^-$. This indicates that $NH_3$ plays an important role in the nitrate formation, but it is now highly in excess in the NCP region so that the nitrate formation is somewhat insensitive to $NH_3$.

Figure 7 presents the dependence of nitrate formation to $NO_2$, $O_3$ and $NH_3$ for the nighttime cases. Again, the results obtained from the three study sites were similar. Nitrate formation was very sensitive to both of $NO_2$ and $O_3$. Adjusting the abundances of $NO_2$ or $O_3$ would lead to almost a linear response in the model-simulated nitrate formation. As discussed above, the nocturnal nitrate formation was mainly controlled by the hydrolysis reactions of $N_2O_5$, which is the product of the reactions of $NO_2$ with $O_3$. In comparison, nitrate formation was not sensitive to $NH_3$ at all three sites. Interestingly, large reductions of $NH_3$ (c.a. >60% at Yucheng and >90% in Ji'nan) would result in a slight increase of the $NO_3^-$ aerosol formation. This should be due to the increase of aerosol acidity by reducing the $NH_3$ levels, which could change the partitioning of the formations of both nitrate and $ClNO_2$ from the $N_2O_5$ hydrolysis. Increasing the aerosol acidity would restrict the reaction of $NO_2^+$ with $Cl^-$ yielding $ClNO_2$, and hence enhance the formation of nitrate aerosol (Roberts et al., 2008). We conducted sensitivity tests without the inputs of $Cl^-$, and the results didn't show any increase in nitrate formation with reduction of $NH_3$ (figures not shown).

It should be noted that the Mt. Tai site is located at around 1465 m a.s.l., which is almost near the top of PBL in summer. Thus the Mt. Tai data can provide insights into the chemical conditions in the top boundary layer during the day and in the residual layer during the night. Our observations at Mt. Tai demonstrate the serious nitrate aerosol pollution throughout the PBL in the NCP region. Furthermore, the nitrate formation mechanisms, including the major formation routes and sensitivities to NOx, $O_3$ and $NH_3$, were fairly consistent between Mt. Tai and the surface sites. This implies the regional homogeneity in the in-situ formation of fine nitrate aerosol within the PBL over the NCP region.

### 3.4. Implications for control policy

The above analyses revealed the important roles of $NO_2$ and $O_3$ in the nitrate formation at three different types of areas. Although $NH_3$ can facilitate the partitioning of $HNO_3$ to the

aerosol phase, it seems that the summertime nitrate formation is less sensitive to $NH_3$ due to the $NH_3$-rich environments in the NCP region. To achieve a comprehensive understanding of the effect of $NH_3$ on nitrate formation, a large set of theoretical simulations were designed with varying initial concentrations of $NO_2$ and $NH_3$. The multi-phase chemical box model was initialized by a typical pollution and meteorological condition in the NCP region (see Table S5 for the detailed modeling setup), and was run to simulate the daytime nitrate formation from 8:00 to 19:00 LT. The initial concentrations of $NO_2$ and $NH_3$ were set to vary in wide ranges of 0-200 ppbv and 0-40 ppbv, to cover a variety of real atmospheric conditions. The dependence of the model-simulated nitrate increment ($\Delta NO_3^-$) to the pair of $NO_2$ and $NH_3$ can be established.

Figure 8 shows the contour plot of the model-simulated daytime $\Delta NO_3^-$ as a function of $NO_2$ and $NH_3$ concentrations. Several interesting aspects are noteworthy from the figure. First, $NH_3$ indeed plays a very important role in prompting the nitrate formation. A relatively small amount of $NH_3$ could significantly enhance the nitrate formation efficiency of $NO_X$. For example, formation of 25 μg m$^{-3}$ of $NO_3^-$ would consume 116 ppbv of $NO_2$ in the absence of $NH_3$, but only need 16 ppbv of $NO_2$ in the presence of 10 ppbv of $NH_3$. Second, at high $NH_3$ conditions (e.g., the right panel of the figure), the nitrate formation becomes insensitive to $NH_3$. Nitrate formation is mainly limited by $NO_2$ when $NH_3$ is in excess. Third, the nitrate formation regimes can be classified into three types, namely, "$NO_X$-limited at $NH_3$-deficient condition", "$NH_3$-controlled", and "$NO_X$-limited at $NH_3$-rich condition", according to the concentration ratios of $NO_2$ and $NH_3$. Identification of the nitrate formation regime is a fundamental step towards the formulation of science-based control policy of nitrate pollution.

Similarly, we also performed theoretical simulations to examine the detailed dependence of nocturnal nitrate formation to both $NO_2$ and $O_3$. The detailed model configuration is given in Table S6. The initial concentrations of $NO_2$ and $O_3$ were set to vary in the range of 0-80 ppbv to represent various nocturnal environments. The contour plot is shown in Figure 9. It clearly shows the three categories of nighttime nitrate formation regimes, i.e., "$NO_X$-limited" under high $O_3$ and low $NO_2$ conditions, "$O_3$-limited" at low $O_3$ and high $NO_2$ conditions, and "mixed-limited" by both $NO_X$ and $O_3$. The ambient pollution conditions measured at the three

study sites in the present study were generally lie in the mixed-limited regime. Effective control measures could be established based upon the diagnosis of the nitrate formation regimes.

Our findings have important implications for the control policy of regional aerosol pollution. Our observations demonstrate the increasing trend and serious situation of nitrate pollution over the NCP region. Given the decline of sulfate and primary particles in the recent decade (Wang et al., 2013), nitrate should be a major target for the future control of haze pollution in China. The observation-based modeling analyses in this study suggest that the summertime nitrate formation in the NCP region is mainly controlled by $NO_X$ and $O_3$ (particularly in nighttime). Recent studies have also confirmed the increasing trends of surface $O_3$ levels in the past decades in several major fast-developing regions of eastern China (Ding et al., 2008; Xu et al., 2008; Wang et al., 2009b; Xue et al., 2014; Sun et al., 2016). Therefore, further reduction of anthropogenic $NO_X$ emissions and mitigation of regional $O_3$ pollution should be an efficient way to alleviate the nitrate-driven haze pollution in China. $NH_3$ also plays a very important role in the nitrate aerosol formation, as a relatively small amount could efficiently prompt the $HNO_3$-to-$NO_3^-$ partitioning and nitrate formation. However, the summertime nitrate formation seems to be less sensitive to $NH_3$ in the NCP region, where ambient $NH_3$ is generally in excess. Indeed, the available field observations of ambient $NH_3$ confirmed the widespread $NH_3$-excess chemical environments in polluted regions of northern China (Meng et al., 2018 and references therein). Thus, it looks like the cutting down the $NO_X$ emissions should be more efficient for the current control of nitrate pollution in the $NH_3$-rich environments. Nevertheless, reduction of $NH_3$ emissions is still very important for the future aerosol pollution control in North China from a long-term perspective, in light of the fact that the nitrate formation would be largely restricted at $NH_3$-poor conditions (see Fig 8).

It is worth noting that in addition to NOx, $O_3$ and $NH_3$, there are also some other factors that influence nitrate formation. For example, VOCs are principal $O_3$ precursors, and regulate the abundances of OH and losses of $NO_3$ (and $N_2O_5$). Thus VOCs can affect the daytime $HNO_3$ formation and nocturnal $N_2O_5$ hydrolysis, which in turn affect the nitrate formation. In

addition, the increasing nitrate aerosol may reduce the $N_2O_5$ uptake and restrict the nocturnal nitrate formation (Chang et al., 2011). The results in Figure 9 only hold if the sensitivity of nitrate production to $N_2O_5$ uptake does not change under different NOx and $O_3$ conditions. Furthermore, the model simulations are constrained to ground-based observations and the chemistry aloft may show a different sensitivity than in Figures 7 and 9. These aspects were not quantified in this study. Further studies are needed to explore the detailed dependence of nitrate formation to the variety of factors including NOx, $O_3$, $NH_3$, VOCs, aerosol composition, and meteorological conditions.

## 4. Conclusions

We report recent field measurements of fine particulate nitrate chemistry at three urban, rural and remote sites in the NCP region. Serious aerosol pollution was observed at all sites, with nitrate accounting for on average 11%-14% of the $PM_{2.5}$. Distinct temporal and spatial distributions of nitrate pollution were found at different sites. The $NO_3^-/PM_{2.5}$ and $NO_3^-/SO_4^{2-}$ ratios have increased significantly in urban Ji'nan over 2005-2015 and at Mt. Tai from 2007 to 2014, highlighting the worsening situation of nitrate pollution in the region. Fine nitrate aerosol was primarily formed via the production of $HNO_3$ followed by its partitioning to the aerosol phase during the day, and by the hydrolysis reactions of $N_2O_5$ on particles during the night. The daytime nitrate formation was mainly controlled by $NO_2$ and to a lesser extent to $O_3$ and $NH_3$, and the nocturnal formation was controlled by both $NO_2$ and $O_3$. $NH_3$ plays a vital role in the nitrate formation by prompting the partitioning of $HNO_3$ to particles. A small amount of $NH_3$ can significantly enhance the efficiency of nitrate formation from $NO_X$. Given the highly $NH_3$-excess condition, the summertime nitrate formation was relatively less sensitive to $NH_3$ in the NCP region. We recommend that further reduction of anthropogenic emissions of $NO_X$ should be the most efficient pathway for the current control of nitrate aerosol, whilst control of regional ozone pollution and $NH_3$ emissions is very important for the future haze pollution control in China. Some recent studies have reported the rapid decrease in the NOx abundances over eastern China since 2011 (Liu et al., 2017). It can be expected that such reduction of NOx would help to alleviate the nitrate particulate pollution in China. More observational studies are needed to further examine the trend in the nitrate

aerosol and assess the contributions of the strict NOx control of China. We also note that the present study was only confined to the summer conditions, and the chemical mix of $NO_2$, $O_3$ and $NH_3$ should be different in wintertime (e.g., higher $NO_X$ levels and relatively low $O_3$ and $NH_3$ concentrations). Further studies are urgently needed to better understand the formation

regimes of nitrate aerosol in winter, when the haze pollution is more serious in China.

**Acknowledgements**

The authors thank Lan Yao, Yanhong Zhu, Junmei Zhang and Hao Wang for their efforts in the field measurements, and thank Dr. Hartmut Herrmann for providing the multi-phase chemical model. This work was funded by the National Key Research and Development

Program of China (No.: 2016YFC0200500), the National Natural Science Foundation of China (No.: 91544213 and 41505111), the Natural Science Foundation of Shandong Province (ZR2014BQ031), the Qilu Youth Talent Program of Shandong University, the Taishan Scholars (ts201712003) and the Jiangsu Collaborative Innovation Center for Climate Change. We also thank the editor and three anonymous reviewers for their comments which are very

helpful for improving the original manuscript.

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

**Table 1.** Statistics (average $\pm$ standard deviation) of the measured aerosol chemical properties, trace gases and meteorological conditions in urban Ji'nan, rural Yucheng, and Mt. Tai

| Site | Ji'nan (Urban) | Ji'nan (Urban) | Yucheng (Rural) | Mt. Tai (Remote) |
|---|---|---|---|---|
| Period | May 2014 | Aug-Sep 2015 | Jun-Jul 2014 | Jul-Aug 2014 |
| $NO_3^-$ ($\mu g\ m^{-3}$) | 8.8±8.2 | 7.4±5.1 | 13.6±10.3 | 6.0±4.6 |
| $SO_4^{2-}$ ($\mu g\ m^{-3}$) | 12.2±7.5 | 12.7±7.9 | 23.6±13.4 | 14.7±8.9 |
| $NH_4^+$ ($\mu g\ m^{-3}$) | 6.8±5.3 | 11.1±8.2 | 11.9±7.7 | 7.3±5.0 |
| $Cl^-$ ($\mu g\ m^{-3}$) | 1.3±2.1 | 1.3±1.7 | 1.2±1.2 | 0.7±0.5 |
| $PM_{2.5}$ ($\mu g\ m^{-3}$) | 68.4±41.7 | 59.3±31.8 | 97.9±53.0 | 50.2±31.7 |
| $NO_3^-/PM_{2.5}$ | 0.12±0.06 | 0.14±0.07 | 0.14±0.07 | 0.11±0.05 |
| $SO_4^{2-}/PM_{2.5}$ | 0.18±0.06 | 0.24±0.10 | 0.27±0.12 | 0.30±0.11 |
| $NH_4^+/PM_{2.5}$ | 0.10±0.05 | 0.18±0.11 | 0.13±0.06 | 0.15±0.07 |
| $Cl^-/PM_{2.5}$ | 0.015±0.016 | 0.024±0.027 | 0.012±0.010 | 0.019±0.025 |
| $[NO_3^-]/[SO_4^{2-}]$ | 1.04±0.46 | 0.98±0.49 | 0.93±0.53 | 0.62±0.33 |
| $NO_2$ (ppb) | 20.5±9.0 | 14.1±4.5 | 16.6±10.7 | 3.0±2.3 |
| $O_3$ (ppb) | 31±19 | 43±36 | 38±26 | 75±21 |
| $SO_2$ (ppb) | 10.4±11.1 | 7.1±4.6 | 4.2±7.4 | 2.4±2.8 |
| CO (ppb) | 1835±2046 | —— | 622±280 | 609±214 |
| Particle diameter (nm) | 41±12 | 55±50 | 78±34 | 66±21 |
| Particle number ($10^3$*# $cm^{-3}$) | 7.1±4.1 | 12±8.3 | 3.0±3.8 | 3.4±2.8 |
| NOR [a] | 0.11±0.07 | 0.16±0.08 | 0.24±0.13 | 0.39±0.20 |
| Excess $NH_4^+$ ($\mu g\ m^{-3}$)[b] | 2.0±2.4 | 4.3±6.4 | 0.9±2.5 | 1.0±1.8 |
| T (°C) | 22.2±4.2 | 23.6±3.4 | 25.4±4.7 | 18.0±2.7 |
| RH (%) | 38.8±19.7 | 66.0±21.0 | 70.3±19.8 | 86.9±12.8 |

[a] NOR (Nitrate Oxidation Ratio) = $[NO_3^-]/([NO_3^-]+[NO_x])$;

[b] Excess $NH_4^+$ = $([NH_4^+]-1.5*[SO_4^{2-}]-[NO_3^-]-[Cl^-])*18$;

5    Note that $[NO_3^-]$, $[NO_x]$, $[NH_4^+]$, $[SO_4^{2-}]$ and $[Cl^-]$ are molar concentrations of $NO_3^-$, $NO_x$, $NH_4^+$, $SO_4^{2-}$ and $Cl^-$, respectively.

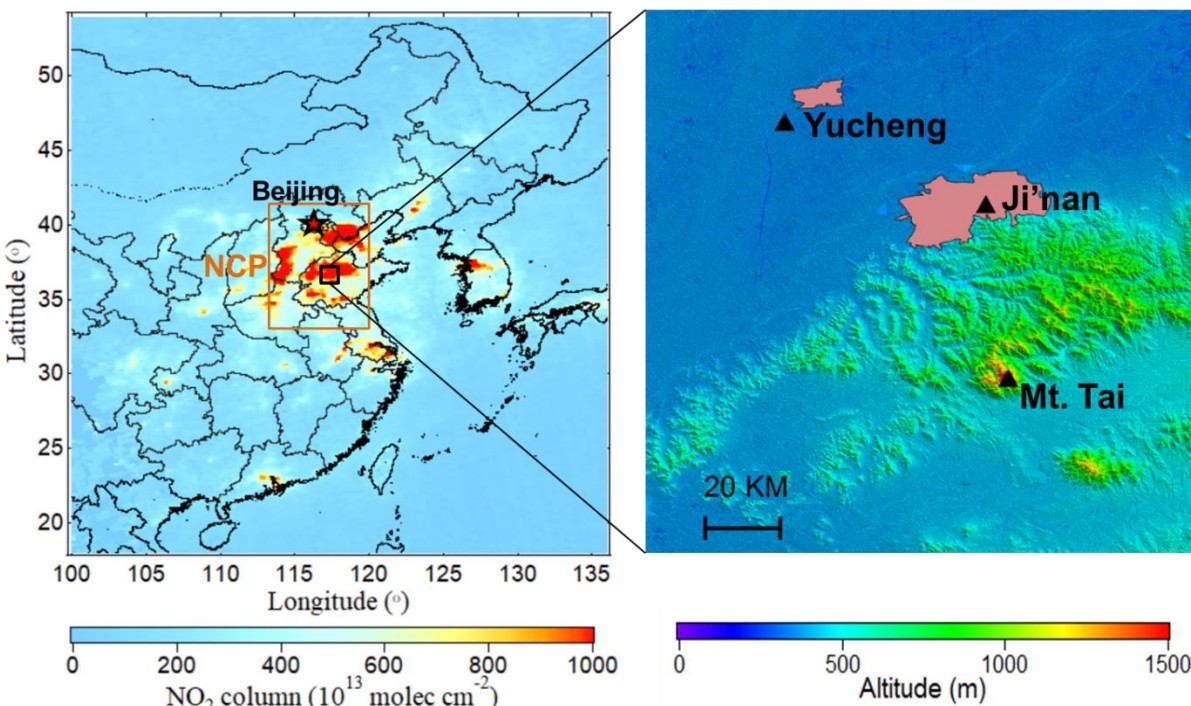

**Figure 1.** Map showing the study region and three measurement sites. The left map is color-coded with the OMI-retrieved tropospheric NO$_2$ column density in July 2014, and the right map is color-coded with the topographic height (the pink regions denote urban areas).

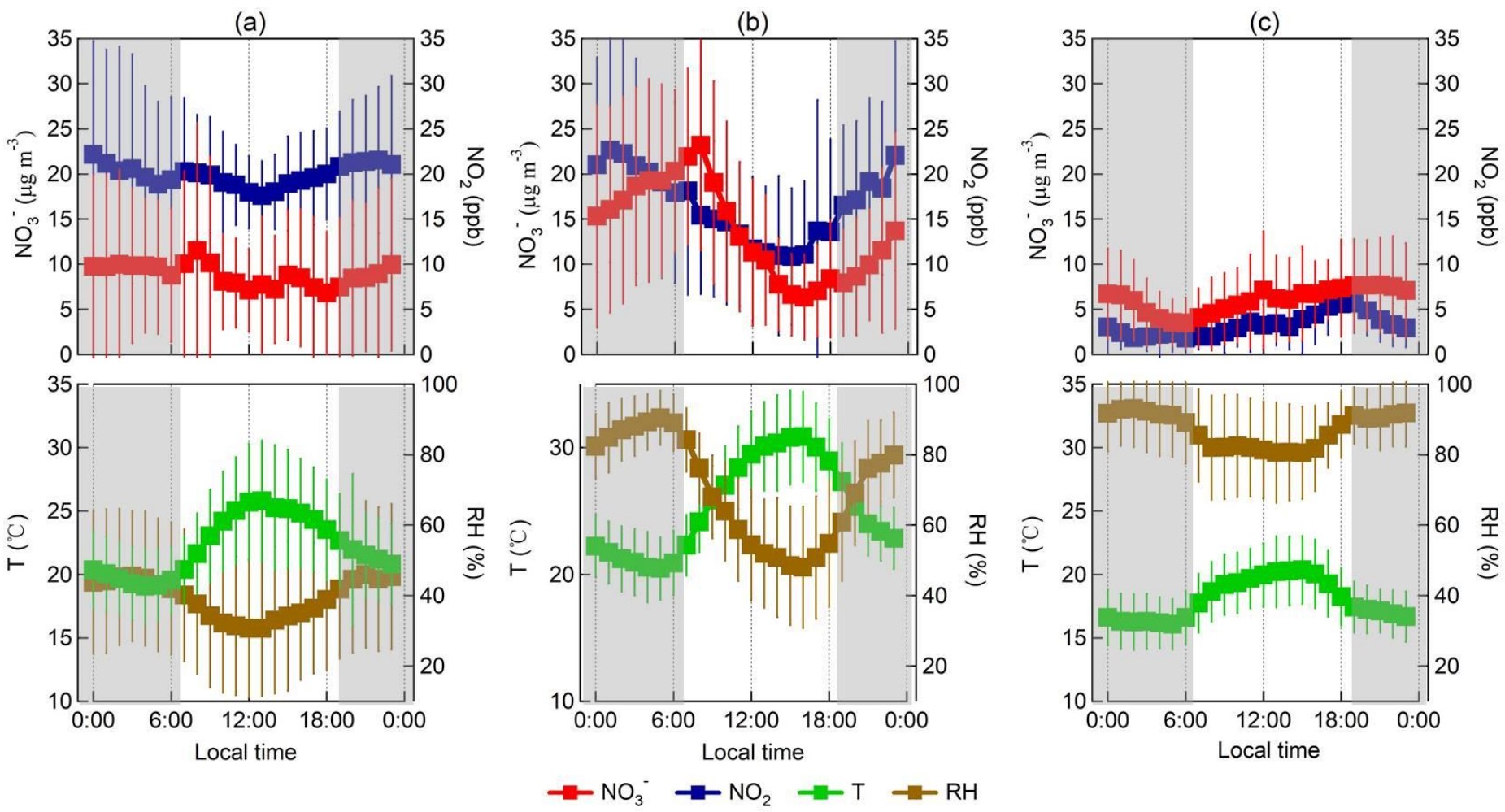

**Figure 2.** Average diurnal variations of fine particulate $NO_3^-$, $NO_2$ and meteorological conditions in (a) urban Ji'nan in May 2014, (b) rural Yucheng, and (c) Mt. Tai. Error bars stand for the standard deviation of the measurements. The shaded area denotes the nighttime period.

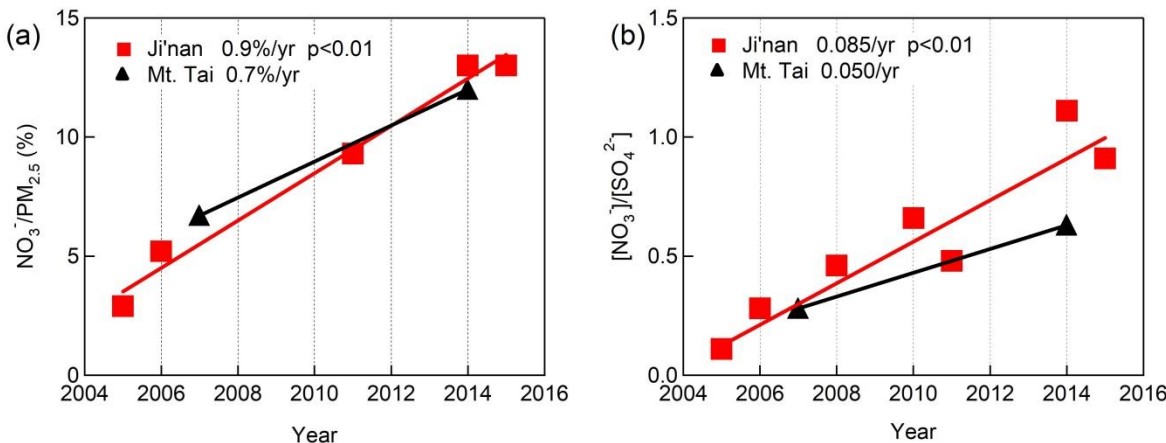

**Figure 3.** Long-term trends of (a) mass ratio of $NO_3^-/PM_{2.5}$ and (b) molar ratio of $NO_3^-/SO_4^{2-}$ in urban Ji'nan and at Mt. Tai in summertime from 2005 to 2015. The fitted lines are derived from the least square linear regression analysis, with the slopes and p values (99% confidence intervals) denoted.

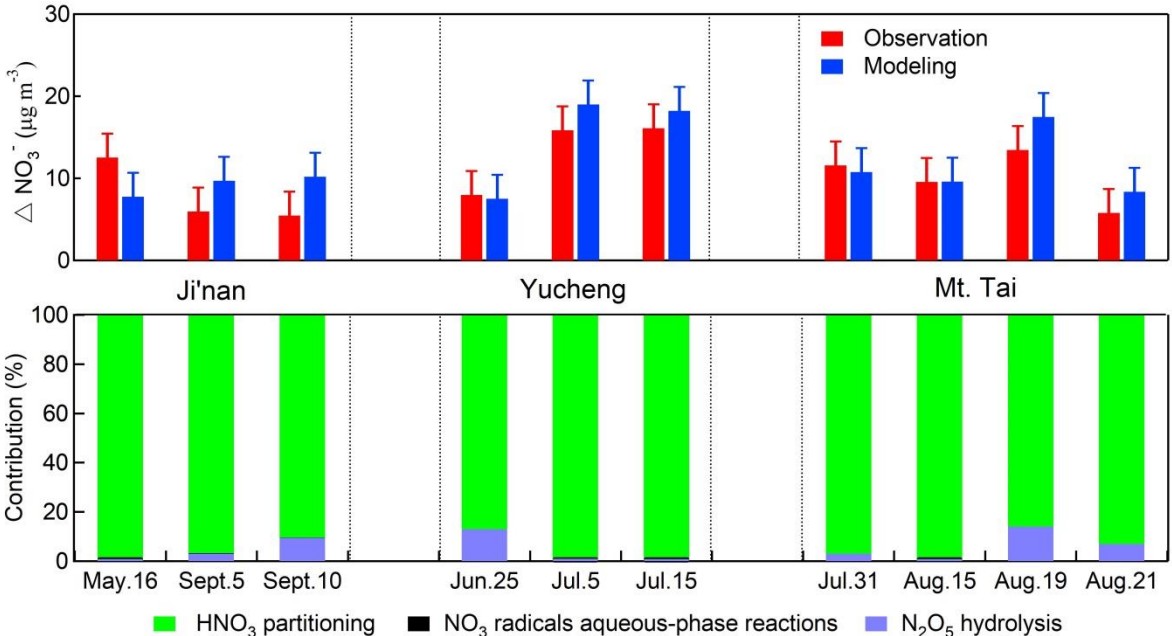

**Figure 4.** Comparison of the model-simulated versus observed nitrate enhancement (upper panel) as well as the contributions from the major three formation pathways (lower panel) for the daytime cases in urban Ji'nan, rural Yucheng and Mt. Tai. The error bars are the standard error of the differences between simulated and observed increase of nitrate aerosol.

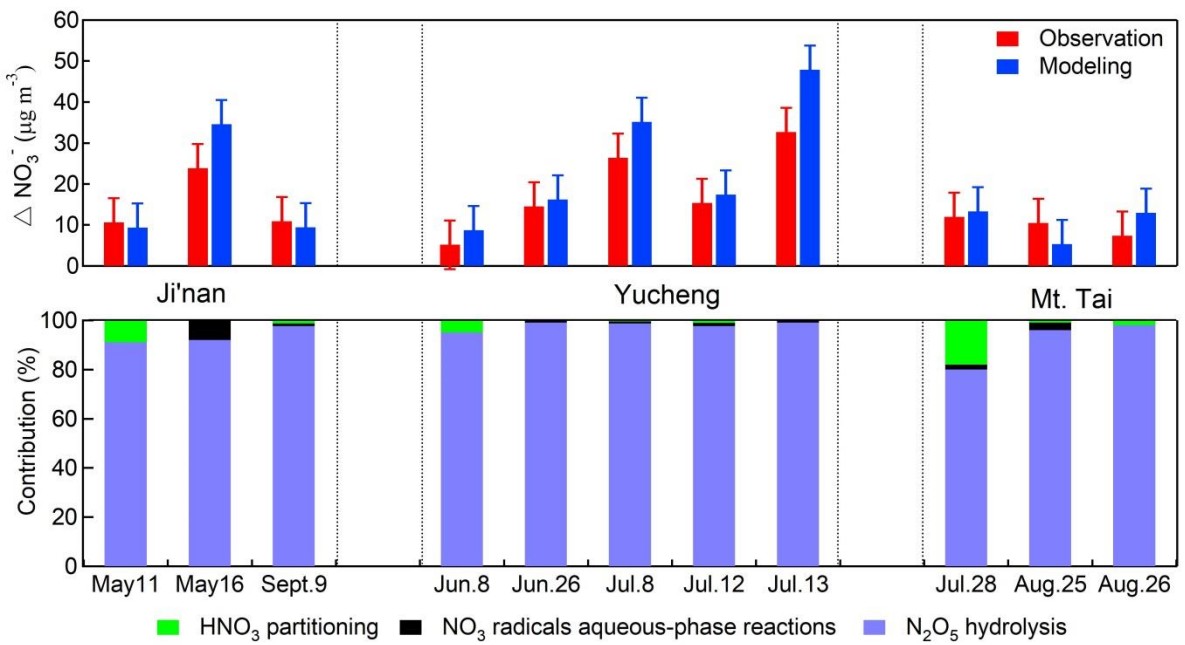

**Figure 5.** The same as Figure 4 but for the selected nocturnal nitrate formation cases.

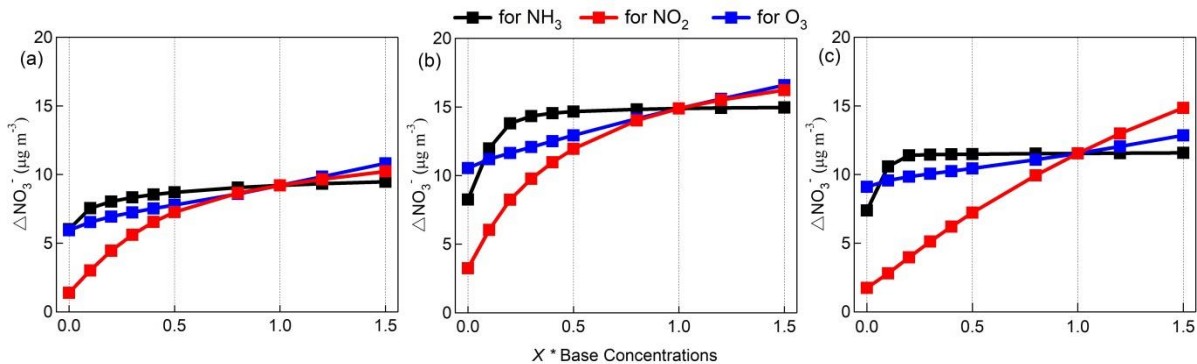

**Figure 6.** Model-simulated daytime average $NO_3^-$ enhancements as a function of the X times of the base concentrations of $NH_3$, $NO_2$ and $O_3$ in (a) urban Ji'nan, (b) rural Yucheng and (c) Mt. Tai. The results are the average of sensitivity analyses for all selected daytime cases.

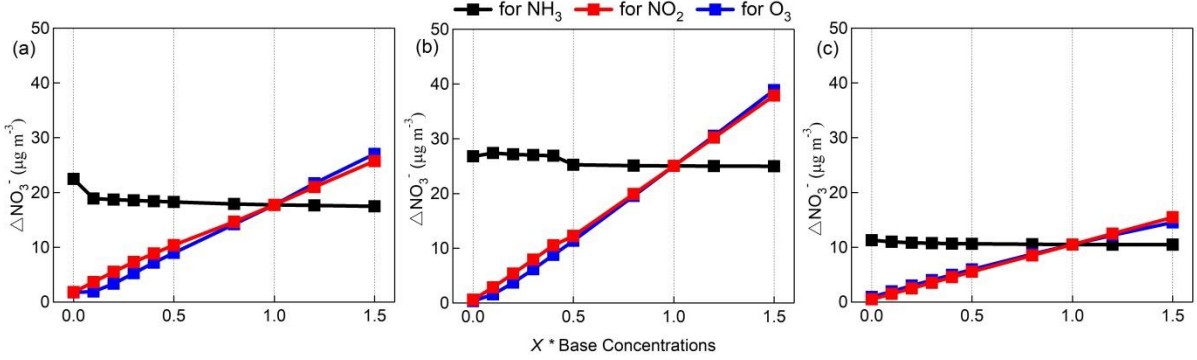

**Figure 7.** The same as Figure 6 but for the nocturnal nitrate formation cases.

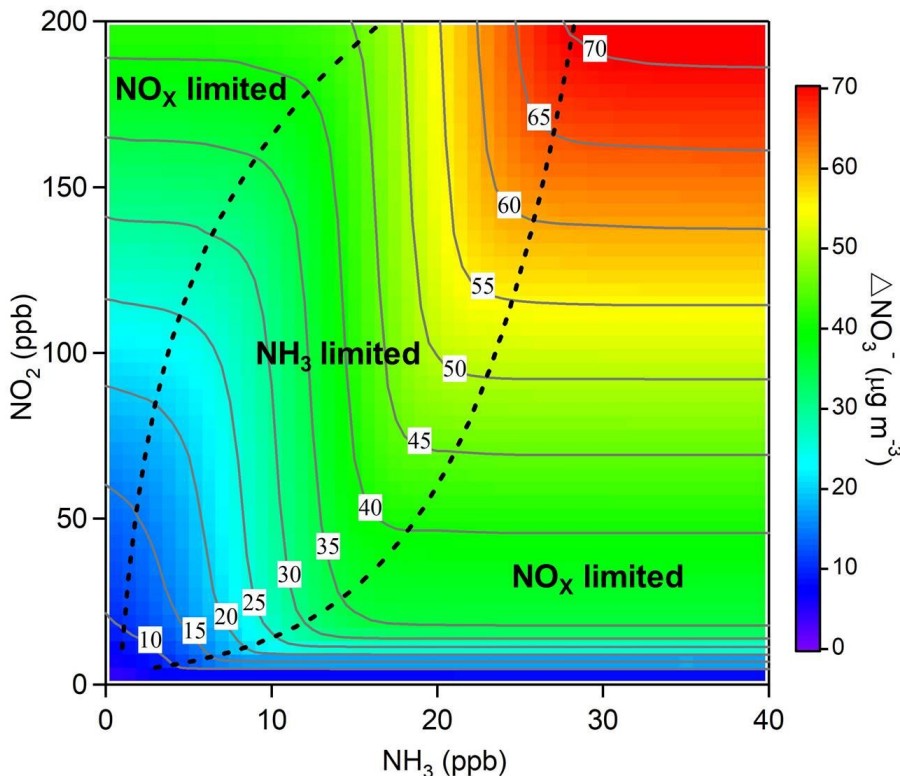

**Figure 8.** Contour plot of the model-simulated daytime $NO_3^-$ formation as a function of the initial concentrations of $NO_2$ (0-200 ppbv) and $NH_3$ (0-40 ppbv). Note that the dashed lines are artificially drawn to separate the three zones with different sensitivity of nitrate formation to $NO_2$ and $NH_3$.

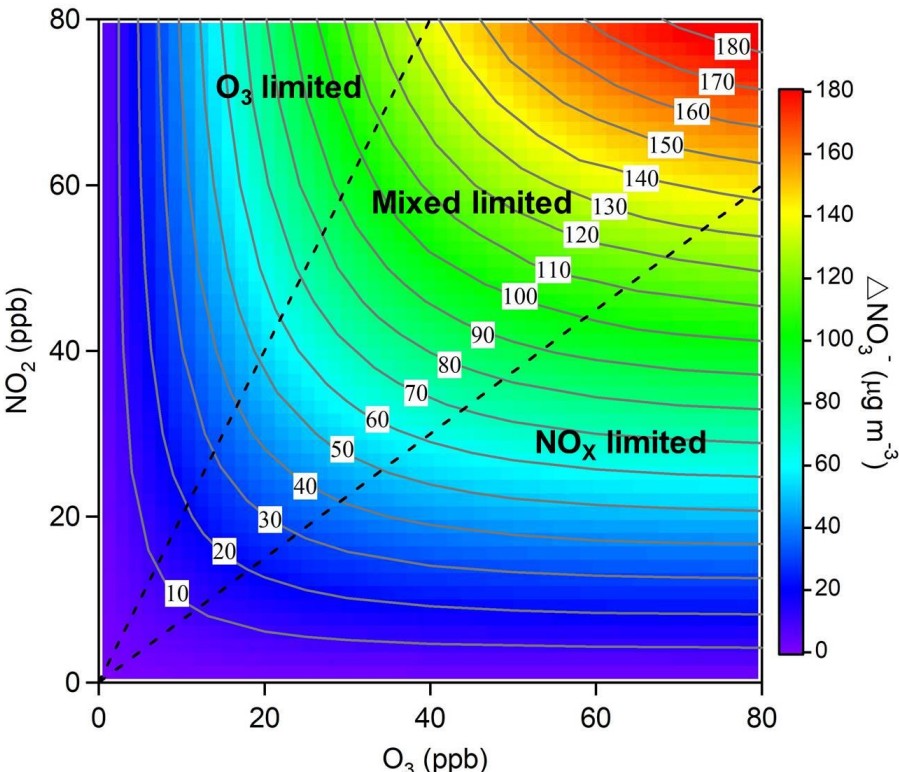

**Figure 9.** Contour plot of the model-simulated nighttime $NO_3^-$ formation as a function of the initial concentrations of $NO_2$ (0-80 ppbv) and $O_3$ (0-80 ppbv). Note that the dashed lines are artificially drawn to separate the three zones with different sensitivity of nitrate formation to $NO_2$ and $O_3$.