# Peer review of "Summertime fine particulate nitrate pollution in the North China Plain: Increasing trends, formation mechanisms, and implications for control policy"

_Atmospheric Chemistry and Physics, 2018_

## Referee Comment (RC1) · Anonymous Referee #1 · 24 Feb 2018

**Review of "Summertime fine particulate nitrate pollution in the North China Plain: Increasing trends, formation mechanisms, and implications for control policy" by Wen, L., et al.**

L. Wen and co-authors present a succinct analysis of recent $PM_{2.5}$ observations and trends at urban, rural, and remote sites in the densely populated North China Plain. Their observations suggest aerosol phase nitrate is becoming an increasingly important component of regional $PM_{2.5}$ and use an observationally informed box model to assess its primary formation pathways during the day and at night. Observations of particulate nitrate, sulfate, and total mass first show that the fraction of nitrate has statistically significantly increased while sulfate has simultaneously decreased. Diurnal patterns are presented to show regional differences in nitrate formation processes. Calculated excess particle-phase ammonium suggests that aerosol nitrate is likely limited by the oxidation of $NO_x$, not emissions of $NH_3$. Box model simulations of select day and nighttime nitrate formation events show that daytime formation is largely due to nitric acid partitioning to the particle phase while nocturnal formation is largely the result of aerosol uptake by $N_2O_5$. Lastly, a large number of simulations were conducted, initialized with varying levels of $NO_2$, $O_3$, and $NH_3$ to test the sensitivity of daytime and nighttime particle nitrate formation to these species. Results suggest that reductions in nitrogen oxide emissions may be the most effective method to reduce nitrate aerosol in Northern China.

The analysis presented here is important to the collective understanding of processes impacting summertime particulate nitrate formation. There are certain areas in this manuscript, however, that require further clarification before publication. The main issue is that further details are required about the box model mechanism and its applicability to daytime processes. Specifically, further details are required to explain how the model treats VOC oxidation by OH and $NO_3$, $N_2O_5$ uptake and reaction product partitioning, as well as the partitioning of $HNO_3$ and reaction with $NH_3$. In addition, consideration of the VOC sensitivity to model results in Section 3.4 should be included. Lastly, additional references should be included throughout the manuscript to provide a stronger context for these results. These and additional comments are provided by page and line number (pg:line) below.

**Major Comments**

3:2-3 – The only direct evidence of pH-dependent $N_2O_5$ uptake has been from laboratory studies. With large discrepancies between uptake trends observed in the field and from laboratory studies, the authors do not have enough evidence to make the claim that increasing acidity can lead to an increase in $N_2O_5$ uptake. If anything, increasing acidity should lead to a decrease in particle phase nitrate as more nitrate partitions to gas-phase $HNO_3$.

3:3-5 – The authors should include additional references to previous studies that have both examined the $NO_x$, $O_3$, $NH_3$ contributions to particle phase nitrate and quantified the reaction pathways of the $NO_3$ radical. This might be a good place to also discuss any expected differences between the extent of nitrate aerosol formation during the summer and winter seasons. Much of the relevant work prior to 2012 is reviewed in Brown and Stutz, 2012. A more recent study by Bassandorj, et al., 2017 and references therein also examine this chemistry during winter. The information included up until this point in the introduction is useful, but more context is required to understand remaining questions surrounding nitrate aerosol formation.

> Brown, S. S., & Stutz, J. (2012). Nighttime radical observations and chemistry. *Chem Soc Rev, 41*(19), 6405-6447. doi:10.1039/c2cs35181a

> Baasandorj, M., Hoch, S. W., Bares, R., Lin, J. C., Brown, S. S., Millet, D. B., Martin, R., Kelly, K., Zarzana, K. J., Whiteman, C. D., Dube, W. P., Tonnesen, G., Jaramillo, I. C., & Sohl, J. (2017). Coupling between Chemical and Meteorological Processes under Persistent Cold-Air Pool Conditions: Evolution of Wintertime PM2.5 Pollution Events and N2O5 Observations in Utah's Salt Lake Valley. *Environ Sci Technol, 51*(11), 5941-5950. doi:10.1021/acs.est.6b06603

Table S1 and Chemical Box Model Description in Main Text –
Provide additional information in the text about how $NH_3$ and $HNO_3$ partitioning are related to each other in this model. Since the model does not include the reaction of $HNO_3 + NH_3$, but rather $HNO_3$ partitioning based on particle acidity, it should be briefly mentioned how $NH_3$ impacts this reactions. In addition, include rate constant information in Table S1. To that point, further details need to be provided about the $N_2O_5 \rightarrow NO_3^- + NO_2^+$ reaction, which represents the uptake of $N_2O_5$ onto aerosol. There are many parameterizations that have been used to quantify this process, but there are also large uncertainties and disagreements with field studies (e.g. Chang, et al., 2011). Since this reaction is a major focus of this manuscript, many more details need to be provided for how it was actually treated in the model. In addition, the authors do not include product partitioning between $HNO_3$ and $ClNO_2$. The formation of $ClNO_2$ could significantly reduce the absolute amount of aerosol nitrate formed by $N_2O_5$ chemistry. Lastly, the authors note that this model has been used previously to simulate *nocturnal* particle nitrate formation. Has this model been also validated for *daytime* formation processes?

> Chang, W. L., Bhave, P. V., Brown, S. S., Riemer, N., Stutz, J., & Dabdub, D. (2011). Heterogeneous Atmospheric Chemistry, Ambient Measurements, and Model Calculations of N2O5: A Review. *Aerosol Science and Technology, 45*(6), 665-695. doi:10.1080/02786826.2010.551672

7:8-12 – Provide further information about the number of VOCs that were included as inputs to this model. Also quantify how 'insensitive' the model was to input VOC concentrations and how these sensitivity studies were conducted. During previous summertime studies, nocturnal $NO_3$ and biogenic VOC concentrations have led to a relatively large $NO_3$ reactivity relative to $N_2O_5$ hydrolysis, which makes the model insensitivity here surprising. The authors need to spend more time evaluating this aspect of the model and discussing how this is similar/different to previous studies.

10:22 – How were day and night defined for the $NO_3$ production case studies? In addition, how did the authors separate events that were likely driven by mixing and transport and not chemical production? For example, morning production periods may be a result of vertical mixing, not chemical production.

10:25-11:11 – Include references to previous studies that have assessed the relative contributions of these different product pathways. This will help place these results in a broader context.

12:21-22 –See comment on uptake pH dependence above. Without additional information about the model parameterization of $N_2O_5$ uptake (see previous comment), it is also difficult to see the relationship between acidity, $N_2O_5$ uptake, and particle nitrate. If anything, a decrease in particle nitrate is expected with increasing acidity, as nitrate partitions to the gas phase. Further discussion about this particular model result is required.

12:29-13:4 – Similar to previous comments, the authors need to include additional evidence of the applicability of this box model to daytime conditions. For example, the authors should include at least one figure showing that the model is able to reproduce the absolute amount of particle nitrate that was observed.

Section 3.4 – The authors need to mention the role of VOCs in both the daytime and nighttime sensitivity studies. The results presented in this section are only valid for constant VOC speciation and absolute values. If either of these change with simultaneous reductions in $NO_x$, $NH_3$, and $O_3$, the daytime abundance of OH would also change as well as the contribution from nocturnal $NO_3$+VOC chemistry. These would alter the results presented in Figures 8 and 9. The authors should address this additional sensitivity by testing a few additional cases with changes in initial VOC concentrations. In addition, there is no discussion about how the changing aerosol composition (i.e. increasing nitrate) is expected to change the contribution from $N_2O_5$ heterogeneous chemistry. More particle nitrate has been shown to reduce $N_2O_5$ uptake and it is unclear how or if this sensitivity is included in the model.

**Typographical and Minor Comments:**

1:14 – Change 'include the downtown' to 'include locations downtown'

1:18 – Change 'have significantly increased' to 'have statistically significantly increased'

1:22 – Change 'at daytime' to 'during the day'. Make this change throughout the entire manuscript (e.g. 2:14, 2:29, 9:14, 12:3, etc.)

1:24 – Reword sentence. Suggest changing to, 'The presence of $NH_3$ contributes to the formation of nitrate aerosol during the day, while decreasing formation at night.

2:2 – Change to 'evidence of a rising trend'

2:9 – Remove 'the' before 'climate change'

2:14 – Point out that policy mitigation strategies will also depend on understanding aerosol composition and sources.

2:15 – Clarify particle phase nitrate vs. gas-phase nitrate radical. i.e. change to 'Particle-phase nitrate ($NO_3^-$) is a principle component…'

2:21-24 – Formation of $NO_3$ and $N_2O_5$ does not only occur at night. Add a sentence clarifying that this process also occurs during the day, but rapid photolysis of $NO_3$ and thermal decomposition of $N_2O_5$ minimize this pathway relative to oxidation of $NO_2$ by the OH radical. Also suggest changing to 'the reaction of $NO_2$ and $O_3$ produce the nitrate radical ($NO_3$), which forms an equilibrium with $N_2O_5$ that can be subsequently taken up onto aerosol to enhance nitrate aerosol.

2:25 – Change 'nitrogen oxides' to '$NO_x$'

2:25-26 – Unclear what the authors mean by 'aqueous transformations of the nitrate radical'. The authors should clarify whether they are referring to $NO_3$ VOC oxidation, which can lead to nitrate containing SOA or direct $NO_3$ uptake onto aerosol.

3:12 – Change 'depositions' to ' deposition'

3:15-17 – Rephrase sentence. Suggest changing to ' In comparison, several recent observational studies have indicated an increasingly important role of aerosol nitrate, which may even dominate summertime haze formation in the NCP'

3:20-23 – Change to 'To the best of our knowledge, there are no previous observational reports of increasing nitrate aerosol over northern China. Long-term measurements are necessary to confirm and quantify this trend, and better understand nitrate formation mechanisms in China.

3:26 – Change 'mountainous' to 'remote' for consistency

3:29 – Change to 'statistically significant'

4:4 – Change to 'increasing trend of nitrate aerosol in Northern China,…'

4:17 – Change 'last' to 'worst'

4:28 – Remove ' due to the closer distance'

5:19 – Specify, was particle phase chloride or HCl measured during this study?

6:16 – Provide the number of chemical reactions in the mechanism to provide the reader with a sense for how explicit daytime VOC degradation is treated.

7:2-3 – Change to 'observed in-situ' and 'available data'

7:4-5 – Was a hygroscopic growth factor applied to the aerosol measurements? If so, how was the growth factor curve determined?

7:25-8:5 – Clarify that the reported values are the campaign average ± the standard deviation. Also specify the different years for the Ji'nan results. Since the measurements we not conducted simultaneously, the authors should also discuss expected differences in the reported averages based on the time of year. Lastly, discuss the potential role of

atmospheric mixing and transport and how these processes could affect the results at each site.

8:6 – Clarify what the authors mean by 'different extent of chemical processing'. For example, are the authors referring to $NO_3$ destruction with fresh NO emissions or air transport allowing more processing time?

8:14-17 – Nitrate fractions of 7-14% don't seem to be particularly large and don't 'elucidate the significance of nitrate aerosol in the haze pollution over eastern China'. Perhaps this argument would be more convincing if the authors cited aerosol nitrate fractions from other locations to put these results in context.

8:22 – What about the role of ammonium chloride in the calculation of excess $NH_4$? The authors could also look at the molar ratio of total $NH_3$ (g) +$NH_4$ (p) to total $NO_3$ (p) + $HNO_3$ (g) to assess the extent of excess ammonium.

8:27-9:3 – What are the proposed reasons for different diurnal profiles in the urban and rural locations? Is there any available information about the role of mixing nitrate formed aloft down to the surface in the morning?

9:3 – Change to 'The absolute nighttime $NO_3$ levels'

9:25 – Was the significance test done at the 95 or 99% level? I.e. is $p < 0.05$ or 0.01? Make sure this is consistent throughout the text and figures.

9:26-27 – Change to 'statistically significant'

10:3 – Change to 'Our observations provide direct evidence of a statistically significant increase of summertime nitrate aerosol…'

10:11 – Clarify what mitigations strategies have been implemented. This sentence makes it sound as if the entire pollution problem has already been mitigated.

10:21 – Change to 'deposition'

Table S2 – Average is typically abbreviated Avg. not Ave.

11:3 – See comment above, unclear If 'aqueous' $NO_3$ reactions are referring to VOC oxidation and condensation or direct $NO_3$ uptake and reaction.

11:25-12:11 – Suggest including an example plot in the supplement of the correlation between observed and modeled nitrate aerosol.

---

## Referee Comment (RC2) · Anonymous Referee #2 · 10 Mar 2018

The manuscript "Summertime fine particulate nitrate pollution in the North China Plain: Increasing trends, formation mechanisms, and implications for control policy" by Liang Wen and Co-Authors presents the results from measurements conducted in three sites in the North China Plane (urban, rural and remote), in the summertime of 2014 and 2015. Mass and composition of inorganic soluble ions of PM2.5 were measured, together with aerosol size distributions, NO, NO2, O3, CO, SO2 concentrations and meteorological parameters. The measurements were compared to previous studies to infer temporal trends of the aerosol nitrate. Additionally, the measurements were com-

pared to the output of the RACM2/CAPRAM2.4 model. The model results were also used to infer the dominant nitrate formation mechanism during the day and at night. Ultimately, the Authors performed a sensitivity analysis, modifying the concentrations of precursor gases (NOx or NH3) in their model to probe which scenario would be the most effective in order to reduce PM2.5 pollution in the area.

The Referee thinks that the paper addresses relevant scientific questions within the scope of ACP, presenting data of interest to the scientific community. However, 1) the abstract should be rephrased and made clearer; 2) Additional references should be included to give proper credit to related work; 3) some of the methods and assumptions used in the paper should be better outlined and clarified; and 4) some of the figures should be improved for a more straightforward interpretation. The Referee recommends publication in ACP after the comments below are properly addressed.

Abstract

The Referee thinks that the abstract should be improved. In the current version, a few long sentences and some confusing passages prevent an efficient understanding of the interesting results of the study. In particular:

Page 1, Line 14: The Referee suggests braking the sentence in two parts. One sentence telling about the measurements and one describing the NCP

Page 1, Line 14-16: the expression ". . . downtown and downwind Ji'nan . . ." can be confusing for the Reader that approaches for the first time the description of the measurements sites. Please reword the sentence to make sure that it is clear that those are two distinct sites and that the urban site is downtown Ji'nan and the rural site is downwind of Ji'nan.

Page 1, Line 24-27: The Referee recommends braking the sentence. One sentence for the day time results and one for the night time results. Additionally, please reword the expression ". . . plays a slightly negative role . . ." The word negative is vague and

a possible source of confusion for the Reader. Consider using "contributes to a slight decrease in nitrate" or similar.

Introduction

Page 2, Line 21: The authors should consider adding a reference to Song, C. H. and G. R. Carmichael (2001). "Gas-particle partitioning of nitric acid modulated by alkaline aerosol." Journal of Atmospheric Chemistry 40(1): 1-22.

Page 2, Line 24: The authors should consider adding a reference to Brown, S. S. and J. Stutz (2012). "Night-time radical observations and chemistry". Chem. Soc. Rev., 41, 6405-6447. doi: 10.1039/c2cs35181a.

Page 2, Line 25: The authors should consider adding a reference to Dentener, F. J. and P. J. Crutzen (1993). "Reaction of N2O5 on Tropospheric Aerosols - Impact on the Global Distributions of Nox, O3, and Oh." Journal of Geophysical Research-Atmospheres 98(D4): 7149-7163.

Material and methods

Page 7, Line 8-11: The Referee strongly suggests that the Authors indicate the VOC average data used. This is an important information that is omitted in the manuscript and without which it is not possible to reproduce the model results.

Page 7, Line 11-12: The Referee strongly suggests that the Authors indicate the range used for the VOC concentrations in the sensitivity test. Additionally, the statement ". . . the nitrate formation was insensitive to the input VOC concentrations." should be quantified.

Results and discussion

In the manuscript, there is no mention of chloride in the aerosol particles. Is it because there was none? The Referee recommends that the Authors add a sentence on the amount of chloride in the particles measured during the study.

Page 8-9, Line 29-2: "... nitrate formation process throughout the nighttime with a NO3- increase of 16.9 ug m-3, ..." it is hard to understand where this number comes from. This is because the nighttime is not clearly defined in the manuscript. The Referee suggests to add a definition of night time (maybe using the solar elevation angle) and to add to figure 2 a visual aid (maybe a shaded area) to visually separate night time and daytime.

Page 11, Line 5-11: It is not clear if the RMA slope is from simulated vs observed or vice versa. I guess it is the former case, but it would be advisable to specify if the model over or under predicts the measurements.

Page 11, Line 19-24: I suggest moving this sentence to the next paragraph. The Reader is left hanging at the end of this sentence that,I feel, is a preamble to the first sentence of next paragraph.

Figures and Tables

The Referee recommends adding an additional table with 3 columns: 1)time of the measurements, 2)location name, and 3)description (urban/rural/remote). This would help the reader navigate the paper more easily.

Table 1: The Referee thinks it would interesting for the Reader if the Authors would add mean values and standard deviations for O3, SO2, CO, mean diameter and mean number, as the Authors state that those data were available. Additionally, adding the values for the ratio of the sum or the inorganic species divided PM2.5 would be a valuable information that would avoid extra work for the reader.

Figure 2: Please specify if those are averages over all period and add the x-axis label.

Figure 3: Please add x-axis label and standard deviation.

Figure 4 and 5: Please add uncertainty bars to the histograms in the top panel.

Figure 8 and 9: Please explain in the caption what are the dashed lines.

Page 2, Line 27 and Page 11, Line 12: I suggest removing "Obviously". It is unnecessary and condescending towards the Reader.

Page 3, Line 28: Please specify that in the notation "nitrate/PM2.5" and "nitrate/sulfate" the Authors is referring to ratios.

Page 3, Line 4 and Page 8, Line 10: I suggest removing "relatively". It is unnecessary unless the Authors are able to specify relatively to what.
* * *

---

## Editor Comment (EC1) · D. Parrish (Editor) · 20 Apr 2018

Editor review of "Summertime fine particulate nitrate pollution in the North China Plain: Increasing trends, formation mechanisms, and implications for control policy" by Liang Wen et al.

MS Number: acp-2018-89

**Summary:**

As discussed by the two referees, this paper presents a useful analysis of particulate nitrate pollution in the North China Plain. I concur in their judgment. In addition to the comments of those referees, I suggest that two additional points be discussed:

1) In the Abstract the authors note: "The nitrate/PM$_{2.5}$ and nitrate/sulfate ratios have significantly increased in Ji'nan (2005-2015) and at Mt. Tai (from 2007 to 2014), indicating the worsening situation of regional nitrate pollution." And likewise "This study provides observational evidence of rising trend of nitrate aerosol …" These statements are necessarily correct only if the absolute concentrations of PM$_{2.5}$ and sulfate have remained constant (or increased). If these two species in the denominator of the two ratios have decreased more rapidly than the ratios themselves, then the regional nitrate pollution may be improving in an absolute (but not relative) sense. A short discussion and clarification of this issue should be included.

2) In the Conclusions the authors "recommend that further reduction of anthropogenic emissions of NO$_x$ should be the most efficient pathway for the current control of nitrate aerosol ..." The data discussed in the paper were collected in 2014 and earlier years. Satellite data (e.g., Liu et al., 2017) suggest that NO$_x$ over the North China Plain was increasing during the period covered by these data, but has been decreasing rapidly since 2014. The authors should briefly discuss the likely impact of this NO$_x$ reduction.

**Reference:**

Liu, F., Beirle, S., Zhang, Q., van der A, R. J., Zheng, B., Tong, D., & He, K. (2017). NOx emission trends over Chinese cities estimated from OMI observations during 2005 to 2015. Atmospheric Chemistry and Physics, 17, 9261–9275.

---

## Referee Comment (RC3) · Anonymous Referee #3 · 21 Apr 2018

Fine particulate nitrate pollution has been found to play more and more important role in haze pollution in China. This paper reports measurement results of nitrate and relevant species at three distinctly different sites in the North China Plain, the most polluted region in eastern China, and interprets the main daytime and nighttime formation mechanisms of nitrate and discusses its implications for air pollution measures in this region. This paper gives very important insights into the formation mechanisms of summertime fine particulate nitrate and into the control policy of haze pollution in China. It was very well organized and written and can be accepted for publication in

ACP as the following points are addressed.

Major points: 1)The difference between the Mt. Tai and ground surface sites as well as its implication need to be highlighted. The Mt. Tai site locates around 1465 m a.s.l., which is almost near the top of planetary boundary layer (PBL) in summer. This site is not only a "remote site" in this region, but also can provide more insights into the different chemical mechanisms inside or above the PBL, or in the nocturnal PBL and the residual layer. These issue need to be sharpen in the data analysis or in the discussions.

2)For the MCM modeling of episodes, the model was run at observational-based mode (OBM). Available measurement data, including nitrate, were used as the model inputs. This method of course could help identify the ongoing chemical processes in the air masses, but it is difficult to trace back to the historical contribution of chemical processes. For example, the observed NH4NO3, already existed as initial condition, could be converted into HNO3 through thermodynamics and further cause an "artificial" mechanism from HNO3 partitioning. Is that possible to do some sensitivity test by removing or reduction the observed nitrate concentration in the MCM OBM? Otherwise, the authors should mention the weakness or uncertainty of the observational-base modelling when they interpret the modeling results.

Minor points:

1)Please use same scale in Y-axis for the comparison of results from different sites, such as Figure 2, Figure 6 and Figure 7. I understand that the authors would like to highlight some peaks in each panel. However, it is more important to make a comparison between different sites.

2)About the trends of nitrate/PM2.5 and nitrate/sulfate in Figure 3, can we also show the trends of nitrate, NO2 and O3 concentration if the data are also available?

3)References of MARGA measurement: Please add some references of measurements based on this instrument, especially those done in the high aerosol loading environment in China.

4)Page 7, Line 1 and Line 12-14. "Mixing layer height" and "boundary layer height", please use consistent words. In addition, the boundary layer height not only "affects dry deposition", the boundary layer height (or mixing layer height) determines the dispersion capacity of air pollutants emitted from ground surface.

5)Page 9, line 4-5. The uplifted PBL: the developed PBL or uplifted PBL height.

---

## Author Comment (AC1) · 14 Jun 2018

**Response to Reviewer 1:**

*L. Wen and co-authors present a succinct analysis of recent $PM_{2.5}$ observations and trends at urban, rural, and remote sites in the densely populated North China Plain. Their observations suggest aerosol phase nitrate is becoming an increasingly important component of regional $PM_{2.5}$ and use an observationally informed box model to assess its primary formation pathways during the day and at night. Observations of particulate nitrate, sulfate, and total mass first show that the fraction of nitrate has statistically significantly increased while sulfate has simultaneously decreased. Diurnal patterns are presented to show regional differences in nitrate formation processes. Calculated excess particle-phase ammonium suggests that aerosol nitrate is likely limited by the oxidation of $NO_x$, not emissions of $NH_3$. Box model simulations of select day and nighttime nitrate formation events show that daytime formation is largely due to nitric acid partitioning to the particle phase while nocturnal formation is largely the result of aerosol uptake by $N_2O_5$. Lastly, a large number of simulations were conducted, initialized with varying levels of $NO_2$, $O_3$, and $NH_3$ to test the sensitivity of daytime and nighttime particle nitrate formation to these species. Results suggest that reductions in nitrogen oxide emissions may be the most effective method to reduce nitrate aerosol in Northern China.*

*The analysis presented here is important to the collective understanding of processes impacting summertime particulate nitrate formation. There are certain areas in this manuscript, however, that require further clarification before publication. The main issue is that further details are required about the box model mechanism and its applicability to daytime processes. Specifically, further details are required to explain how the model treats VOC oxidation by OH and $NO_3$, $N_2O_5$ uptake and reaction product partitioning, as well as the partitioning of $HNO_3$ and reaction with $NH_3$. In addition, consideration of the VOC sensitivity to model results in Section 3.4 should be included. Lastly, additional references should be included throughout the manuscript to provide a stronger context for these results. These and additional comments are provided by page and line number (pg:line) below.*

**Response:** we thank the reviewer for the thoughtful review and constructive comments. All of these comments and suggestions are very helpful for improving our manuscript. We have carefully considered and tried to address all of these comments, and significantly revised the manuscript. Briefly, more details about the model mechanism, validation and sensitivity tests have been provided. More references that are relevant to this study have been acknowledged. More discussions of the observational and modelling results have been added. Below we reply in details to the individual comments. For clarity, the reviewer's comments are listed in black italics, while our responses and changes in the manuscript are highlighted in blue and red, respectively.

*Major Comments*

*3:2-3 – The only direct evidence of pH-dependent $N_2O_5$ uptake has been from laboratory studies. With large discrepancies between uptake trends observed in the field and from laboratory studies, the authors do not have enough evidence to make the claim that increasing acidity can lead to an increase in $N_2O_5$ uptake. If anything, increasing acidity*

*should lead to a decrease in particle phase nitrate as more nitrate partitions to gas-phase HNO$_3$.*

**Response:** we agree with the referee that we don't have enough evidence for the dependence of N$_2$O$_5$ uptake to the aerosol acidity, given the large discrepancy between field studies and laboratory efforts. The original statement has been removed from the revised manuscript.

*3:3-5 – The authors should include additional references to previous studies that have both examined the NO$_x$, O$_3$, NH$_3$ contributions to particle phase nitrate and quantified the reaction pathways of the NO$_3$ radical. This might be a good place to also discuss any expected differences between the extent of nitrate aerosol formation during the summer and winter seasons. Much of the relevant work prior to 2012 is reviewed in Brown and Stutz, 2012. A more recent study by Bassandorj, et al., 2017 and references therein also examine this chemistry during winter. The information included up until this point in the introduction is useful, but more context is required to understand remaining questions surrounding nitrate aerosol formation.*

*Brown, S. S., & Stutz, J. (2012). Nighttime radical observations and chemistry. Chem Soc Rev, 41(19), 6405-6447. doi:10.1039/c2cs35181a*

*Baasandorj, M., Hoch, S. W., Bares, R., Lin, J. C., Brown, S. S., Millet, D. B., Martin, R., Kelly, K., Zarzana, K. J., Whiteman, C. D., Dube, W. P., Tonnesen, G., Jaramillo, I. C., & Sohl, J. (2017). Coupling between Chemical and Meteorological Processes under Persistent Cold-Air Pool Conditions: Evolution of Wintertime PM$_{2.5}$ Pollution Events and N$_2$O$_5$ Observations in Utah's Salt Lake Valley. Environ Sci Technol, 51(11), 5941-5950. doi:10.1021/acs.est.6b06603*

**Response:** thanks for the suggestion. Indeed, there are some remaining questions surrounding the nitrate formation mechanisms, such as the highly variable uptake coefficient of N$_2$O$_5$ on particles (γN$_2$O$_5$), the reaction pathways of the NO$_3$ radical and its competition with the N$_2$O$_5$ hydrolysis, the seasonal dependence of the N$_2$O$_5$ hydrolysis reactions, and the vertical mixing of air aloft in the residual layer, etc. We have added the following discussion about this in the revised manuscript.

"To date the detailed relationship between nitrate formation and the chemical mix of NO$_x$, O$_3$ and NH$_3$ is still poorly understood. Field measurement studies have shown that the uptake coefficient of N$_2$O$_5$ onto particles (γN$_2$O$_5$) is highly variable and disagrees with the laboratory-derived parameterizations (Brown and Stutz, 2012; and references therein). The contribution of N$_2$O$_5$ hydrolysis pathway tends to show a seasonal dependence with the largest influence in the winter season (Brown and Stutz, 2012; Baasanforj et al., 2017). Vertical mixing of air aloft in the residual layer may also contribute to the surface nitrate particles (Baasanforj et al., 2017). Consequently, there are still some remaining questions for better understanding the nitrate formation mechanisms."

*Table S1 and Chemical Box Model Description in Main Text –*

*Provide additional information in the text about how NH$_3$ and HNO$_3$ partitioning are related to each other in this model. Since the model does not include the reaction of HNO$_3$ + NH$_3$,*

*but rather HNO₃ partitioning based on particle acidity, it should be briefly mentioned how NH₃ impacts this reactions. In addition, include rate constant information in Table S1. To that point, further details need to be provided about the $N_2O_5 \rightarrow NO_3^- + NO_2^+$ reaction, which represents the uptake of $N_2O_5$ onto aerosol. There are many parameterizations that have been used to quantify this process, but there are also large uncertainties and disagreements with field studies (e.g. Chang, et al., 2011). Since this reaction is a major focus of this manuscript, many more details need to be provided for how it was actually treated in the model. In addition, the authors do not include product partitioning between HNO₃ and ClNO₂. The formation of ClNO₂ could significantly reduce the absolute amount of aerosol nitrate formed by $N_2O_5$ chemistry. Lastly, the authors note that this model has been used previously to simulate nocturnal particle nitrate formation. Has this model been also validated for daytime formation processes?*

*Chang, W. L., Bhave, P. V., Brown, S. S., Riemer, N., Stutz, J., & Dabdub, D. (2011). Heterogeneous Atmospheric Chemistry, Ambient Measurements, and Model Calculations of $N_2O_5$: A Review. Aerosol Science and Technology, 45(6), 665-695. doi:10.1080/02786826.2010.551672*

**Response:** the chemical box model we used in the present study is a little bit different from the commonly used models which usually adopt the experiment-derived parameterizations to represent heterogeneous processes. Our model explicitly describes the gas-phase reactions (by RACM2), aqueous-phase reactions (by CAPRAM2.4), and the gas-aqueous partitioning (phase transfer) processes. We are sorry that the original Table S1 and related descriptions missed some important information about the model configuration. In the revised manuscript, we have elaborated more about the details of representation of some key chemical processes in the model. Below we briefly reply to the specific comments of the reviewer.

**(1) On the representations of the HNO₃ and NH₃ reactions:**

Indeed, the model does not include the reaction of $HNO_3+NH_3=NH_4NO_3$. It describes the gas-to-aqueous phase partitioning and the aqueous phase reactions of HNO₃ and NH₃ by the following reactions.

$$HNO_3(g) \leftrightarrow HNO_3(a) \quad K_{1f}, K_{1b} \qquad (R1)$$

$$HNO_3(a) \leftrightarrow H^+ + NO_3^- \quad K_{2f}, K_{2b} \qquad (R2)$$

$$NH_3(g) \leftrightarrow NH_3(a) \quad K_{3f}, K_{3b} \qquad (R3)$$

$$NH_3(a) + H_2O(a) \leftrightarrow NH_4^+ + OH^- \quad K_{4f}, K_{4b} \qquad (R4)$$

$$H^+ + OH^- \leftrightarrow H_2O(a) \quad K_{5f}, K_{5b} \qquad (R5)$$

Reactions (R1) and (R3) describe the partitioning of HNO₃ and NH₃ between the gas and aqueous phases, with $K_{1f}$, $K_{1b}$, $K_{3f}$ and $K_{3b}$ are functions of the molecular speeds, gas-phase diffusion coefficients, accommodation coefficients, and Henry coefficients of HNO₃ and NH₃. Reactions (R2) and (R4) represent the reversible ionization equilibrium of HNO₃ and NH₃ in the aqueous phase. Reaction (R5) links the partitioning of HNO₃ and NH₃ with each other. Briefly, increasing NH₃ would decrease the aerosol acidity (by providing more OH⁻), which

would then enhance the partitioning of $HNO_3$ to the aqueous phase as well as formation of $NO_3^-$.

In the revised manuscript, Table S1 has been revised to include the above detailed description of the $HNO_3$ and $NH_3$ partitioning processes. The rate constants for all of the reactions have been provided in Table S1. The following statements have been also added to the revised manuscript to elaborate this reaction pathway.

"This model explicitly describes the gas-to-aqueous phase partitioning of various chemical species, which connects the detailed chemical reactions in both gas and aqueous phases. The chemical reactions representing the nitrate formation in the model are outlined in Table S1. Briefly, these reactions can be categorized into three major formation pathways, namely, partitioning of gaseous $HNO_3$ to the aerosol phase, hydrolysis reactions of $N_2O_5$, and aqueous phase reactions of $NO_3$ radicals. The $HNO_3$ partitioning is largely affected by the availability of $NH_3$, since the partitioning of $NH_3$ would decrease the aerosol acidity and hence enhance the partitioning of $HNO_3$ to the aerosol phase (see Table S1)."

**(2) On the representation of the $N_2O_5$ hydrolysis process**

Similar to the $HNO_3$ partitioning, the model describes explicitly the gas-to-aqueous phase partitioning of $N_2O_5$ as well. The uptake coefficient of $N_2O_5$ on particles ($\gamma N_2O_5$) was not parameterized within the model. This heterogeneous process is represented by the following chemical reactions in the model.

$$N_2O_5\,(g) \leftrightarrow N_2O_5\,(a) \qquad K_{6f},\ K_{6b} \qquad\qquad (R6)$$

$$N_2O_5(a) + H_2O(a) \rightarrow 2H^+ + 2NO_3^- \qquad k_7 \qquad\qquad (R7)$$

$$N_2O_5(a) \rightarrow NO_2^+ + NO_3^- \qquad k_8 \qquad\qquad (R8)$$

$$NO_2^+ + H_2O(a) \rightarrow 2H^+ + NO_3^- \qquad k_9 \qquad\qquad (R9)$$

$$NO_2^+ + Cl^- \rightarrow ClNO_2\,(a) \qquad k_{10} \qquad\qquad (R10)$$

The reaction (R6) describes the partitioning of $N_2O_5$ between the gas and aqueous phases, with $K_{6f}$ and $K_{6b}$ being functions of the molecular speeds, gas-phase diffusion coefficients, accommodation coefficients, and Henry coefficients of $N_2O_5$. Reactions (R7)-(R9) describe the aqueous-phase reactions of $N_2O_5$ with liquid water forming nitrate, with the reaction (R8) being the fastest reaction pathway. Reaction (R10) presents the formation of $ClNO_2$ from the $N_2O_5$ hydrolysis.

In the revised manuscript, we have clearly elaborated about the treatment of $N_2O_5$ hydrolysis process by providing the above detailed chemical reactions in Table S1 and also adding the following statements in the main text.

"For the $N_2O_5$ hydrolysis process, the uptake coefficient of $N_2O_5$ on particle surfaces ($\gamma N_2O_5$) is the parameter with large uncertainty in modeling studies. Recent studies have shown that $\gamma N_2O_5$ tends to be largely variable and significant discrepancy exists between field-derived laboratory-derived parameterizations (Chang et al., 2011). The RACM/CAPRAM model used in this study doesn't take $\gamma N_2O_5$ into account but describes explicitly the $N_2O_5$ gas-to-aqueous

phase partitioning as well as its subsequent aqueous phase reactions. See Table S1 for the detailed treatment of the $N_2O_5$ hydrolysis processes in the model."

**(3) On the applicability of the model to simulation of daytime nitrate formation**

To our knowledge, this multi-phase chemical box model has not been applied to simulate the daytime nitrate formation in previous studies. However, the model worked quite well for reproducing the observed nitrate increase for the selected cases at three study sites in the present study. Figures 4 and 5 clearly show the comparison between modeled and observed nitrate increase for the selected 21 daytime and nighttime cases, and the scatter plots of the modeled versus observed nitrate increments for the daytime and nighttime cases are shown in Figure R1. Besides, we also compared the modelling results against observations for the individual cases. Figure R2 shows the time series of observed vs. modeled nitrate and related species for two typical cases at daytime and nighttime, respectively. Overall, these figures clearly show the applicability of the box model to the simulation of daytime (and nighttime) nitrate formation. These figures have been provided in the revised supplementary materials.

[Figure]

**Figure R1.** Scatter plots of the modeled versus observed increase of particulate nitrate for the selected daytime (a) and nighttime (b) cases

[Figure]

**Figure R2.** Comparison of modeled versus observed nitrate concentrations as well as related species for two typical cases at (a) daytime and (b) nighttime

*7:8-12 – Provide further information about the number of VOCs that were included as inputs to this model. Also quantify how 'insensitive' the model was to input VOC concentrations and how these sensitivity studies were conducted. During previous summertime studies, nocturnal $NO_3$ and biogenic VOC concentrations have led to a relatively large $NO_3$ reactivity relative to $N_2O_5$ hydrolysis, which makes the model insensitivity here surprising. The authors need to spend more time evaluating this aspect of the model and discussing how this is similar/different to previous studies.*

**Response:** over 40 VOC species were considered in the modeling analyses in the present study. The detailed VOC compounds and their concentrations as the model inputs have been documented in a table in the revised supplement.

We should note that we didn't have VOC measurements during the present study, and we only took the campaign-average concentrations of VOCs available from previous studies for the same study sites (or study area) as the model inputs. The model was initialized with such average VOC concentrations. Sensitivity studies were conducted by adjusting the initial VOC concentrations to 0.5 or 1.5 times of the base data, and the model-simulated nitrate increases were compared between the sensitivity tests and base runs. As shown from Figure R3, both sensitivity model runs produced comparable daytime and nocturnal nitrate formation to the base runs (the differences were within 12%). This should be mainly due to the low levels of biogenic VOCs (i.e., isoprene and pinenes) at the study sites, and the reactions of $NO_3$ with BVOCs may only account for a small fraction of the total $N_2O_5$ loss.

In the revised manuscript, the original statements have been revised as follows to discuss this aspect, with Figure R3 being added in the supplement.

"The VOC measurements were not made during the present study, and we used the campaign average data previously collected in the same areas during summertime for approximation (Zhu et al., 2016 and 2017). The detailed VOC species and their concentrations as the model input are documented in Table S3. We conducted sensitivity tests with 0.5 or 1.5 times of the initial VOC concentrations, and found that the model simulation was somewhat insensitive to the initial VOC data (the differences between sensitivity tests and base run were within 12%; see Figure S1). This should be mainly due to the low levels of biogenic VOCs in the study area. Given the lack of in-situ VOC measurements, however, the treatment of VOC data presents a major uncertainty in the presented modeling analyses."

[Figure]

**Figure R3.** Sensitivity of the model-simulated (a) daytime and (b) nighttime nitrate formation to the initial VOCs

*10:22 – How were day and night defined for the NO₃ production case studies? In addition, how did the authors separate events that were likely driven by mixing and transport and not chemical production? For example, morning production periods may be a result of vertical mixing, not chemical production.*

**Response:** in the present study, the day and night time windows were defined as 7:00-19:00 and 19:00-07:00 local time, respectively. The selected nitrate formation cases should meet the following criteria: 1) the nitrate accumulation process should last for a considerable time period; 2) the observed NOR (NOR=$[NO_3^-]/([NO_3^-]+[NO_x])$) was increasing throughout the event; (3) the meteorological conditions were stable with constant small winds or a calm condition, without wet deposition. These criteria ensure that the observed nitrate formation was confined to the same air mass. To avoid the potential influence of vertical mixing in the early morning, the data in 06:00-09:00 local time at the surface sites (Ji'nan and Yucheng) have been excluded from the revised analyses. The following statements have been added in the revised manuscript to elaborate more about this issue, and the relevant discussions have been updated throughout the manuscript.

The selected cases met the following criteria: 1) the nitrate formation (accumulation) process should last for a considerable time period (i.e., at least three hours); 2) the observed NOR (NOR=$[NO_3^-]/([NO_3^-]+[NO_x])$) was increasing throughout the event; 3) the meteorological conditions were stable with constant wind direction or a calm condition and without wet deposition; 4) the data in the early morning period (i.e., 06:00-09:00 LT) were excluded from analyses to eliminate the potential influence from downward mixing of air aloft to the surface sites.

*10:25-11:11 – Include references to previous studies that have assessed the relative contributions of these different product pathways. This will help place these results in a broader context.*

**Response:** this suggestion has been adopted in the revised manuscript. The following discussion has been added to compare our results to the related previous studies.

"These results are in line with the previous studies that have assessed the nitrate formation pathways. For example, Pathak et al. (2011) found that the $N_2O_5$ hydrolysis contributed to 50%-100% of the nocturnal nitrate formation in Beijing and Shanghai. Based on the field measurements of $N_2O_5$ and related parameters, Wang et al. (2017) suggested that the $N_2O_5$ hydrolysis contributed comparably to or even higher than the partitioning of $HNO_3$ to nitrate formation in Beijing in a daily basis. Overall, the significant roles of $HNO_3$ partitioning and $N_2O_5$ hydrolysis in nitrate formation have been well outlined (Brown and Stutz, 2012)."

Pathak, R. K., Wang, T., and Wu, W. S.: Nighttime enhancement of $PM_{2.5}$ nitrate in ammonia-poor atmospheric conditions in Beijing and Shanghai: Plausible contributions of heterogeneous hydrolysis of $N_2O_5$ and $HNO_3$ partitioning, Atmos. Environ., 45, 1183-1191, 2011.

Wang, H., Lu, K., Chen, X., Zhu, Q., Chen, Q., Guo, S., Jiang, M., Li, X., Shang, D., Tan, Z., Wu, Y., Wu, Z., Zou, Q, Zheng, Y., Zeng, L., Zhu, T., Hu, M., and Zhang, Y.: High $N_2O_5$ concentrations observed in urban Beijing: implications of a large nitrate formation pathway,

Environ. Sci. Tech., 4, 416-420, 2017.

Brown, S. S. and Stutz, J.: Nighttime radical observations and chemistry, Chem. Soc. Rev., 41, 6405-6447. doi: 10.1039/c2cs35181a, 2012.

*12:21-22 –See comment on uptake pH dependence above. Without additional information about the model parameterization of $N_2O_5$ uptake (see previous comment), it is also difficult to see the relationship between acidity, $N_2O_5$ uptake, and particle nitrate. If anything, a decrease in particle nitrate is expected with increasing acidity, as nitrate partitions to the gas phase. Further discussion about this particular model result is required.*

**Response:** the model representation of the $N_2O_5$ hydrolysis process has been described above and provided in the revised manuscript. We agree with the reviewer that there is no enough evidence for the dependence of $N_2O_5$ uptake to aerosol acidity. As shown from the revised Fig. 7 (see below), the model-simulated nocturnal nitrate formation is quite insensitive to the abundance of $NH_3$, although large reductions of $NH_3$ resulted in slight increases of nitrate at Ji'nan and Yucheng. We have checked for this result by examining all of the reaction rates related to nitrate formation, and found that it may be due to the change in the partitioning of formations of nitrate and $ClNO_2$ from the $N_2O_5$ hydrolysis. Increasing the aerosol acidity would restrict the reaction of $NO_2^+$ with $Cl^-$ producing $ClNO_2$ (since $Cl^-$ reacts with $H^+$ more quickly), and thus would enhance the formation of nitrate aerosol through reaction (R9). We conducted sensitivity tests without the inputs of $Cl^-$, and the results showed that the nighttime nitrate formation is insensitive to $NH_3$ (see Fig. R4).

In the revised manuscript, the original statements have been revised as follows.

"In comparison, nitrate formation was not sensitive to $NH_3$ at all three sites. Interestingly, large reductions of $NH_3$ (c.a. >60% at Yucheng and >90% in Ji'nan) would result in a slight increase of the $NO_3^-$ aerosol formation. This should be due to the increase of aerosol acidity by reducing the $NH_3$ levels, which could change the partitioning of the formations of both nitrate and $ClNO_2$ from the $N_2O_5$ hydrolysis. Increasing the aerosol acidity would restrict the reaction of $NO_2^+$ with $Cl^-$ yielding $ClNO_2$, and hence enhance the formation of nitrate aerosol. We conducted sensitivity tests without the inputs of $Cl^-$, and the results didn't show any increase in nitrate formation with reduction of $NH_3$ (figures not shown)."

[Figure]

**Revised Figure 7.** Model-simulated nighttime average $NO_3^-$ enhancements as a function of the X times of the base concentrations of $NH_3$, $NO_2$ and $O_3$ in (a) urban Ji'nan, (b) rural Yucheng and (c) Mt. Tai.

[Figure]

**Figure R4.** The same as above but without the inputs of Cl$^-$ data in the model.

*12:29-13:4 – Similar to previous comments, the authors need to include additional evidence of the applicability of this box model to daytime conditions. For example, the authors should include at least one figure showing that the model is able to reproduce the absolute amount of particle nitrate that was observed.*

**Response:** as discussed above, the RACM2/CRPRAM2.4 multi-phase model overall worked well for the simulation of nitrate formation during the day. The model reasonably reproduced the observed nitrate formation for the selected cases in the present study. Some evidence including the scatter plots of modeled versus observed nitrate increase as well as time series for typical cases have been provided in the revised supplement. See the response to the above comment for the details (including Figures R1 and R2).

*Section 3.4 – The authors need to mention the role of VOCs in both the daytime and nighttime sensitivity studies. The results presented in this section are only valid for constant VOC speciation and absolute values. If either of these changes with simultaneous reductions in $NO_x$, $NH_3$, and $O_3$, the daytime abundance of OH would also change as well as the contribution from nocturnal $NO_3+VOC$ chemistry. These would alter the results presented in Figures 8 and 9. The authors should address this additional sensitivity by testing a few additional cases with changes in initial VOC concentrations. In addition, there is no discussion about how the changing aerosol composition (i.e. increasing nitrate) is expected to change the contribution from $N_2O_5$ heterogeneous chemistry. More particle nitrate has been shown to reduce $N_2O_5$ uptake and it is unclear how or if this sensitivity is included in the model.*

**Response:** we agree with the reviewer that VOCs indeed play an important role in the nitrate formation. VOCs are principal ozone precursors, and regulate the daytime abundances of OH and nocturnal loss of $NO_3$ (and $N_2O_5$). Thus VOCs can affect the formation of $HNO_3$ during the day and the $N_2O_5$ reactivity at night, both of which in turn affect the nitrate formation.

There are many factors that can influence the nitrate formation, such as NOx, $O_3$, $NH_3$, VOC speciation and abundances, and aerosol compositions. The detailed dependence of nitrate formation to all of these factors is very complex. In this study, we chose to only examine the dependence of nitrate formation to NOx, $O_3$ and $NH_3$, with constant VOC levels and speciation. The average VOC concentrations previously collected at Mt. Tai were used to initialize the model to represent the regional average condition for VOCs.

Although we don't investigate the dependence of nitrate formation to VOCs, we think that changing VOCs should not **qualitatively** change the results presented in Figs. 8 and 9 (the relationship of nitrate with $NO_2$, $O_3$ and $NH_3$). During the day, both $NO_2$ and VOCs affect the formation of $HNO_3$, and $NO_2$ may be more important because it is the direct precursor of $HNO_3$. $NH_3$ does not affect the $HNO_3$ formation but enhances its partitioning to the aerosol phase. Thus VOCs should not alter the relationship of nitrate with $NO_2$ and $NH_3$ as described in Fig. 8, as $NO_2$ and $NH_3$ actually contribute to nitrate formation in different manners. For the nocturnal formation, both $NO_2$ and $O_3$ are direct precursors of $N_2O_5$, while VOCs only affect nitrate formation indirectly by altering the budget of $N_2O_5$ loss via the BVOCs+$NO_3$ reactions. Hence VOCs should also not qualitatively change the relationship of nitrate with $NO_2$ and $O_3$ as shown in Fig. 9.

Furthermore, we have conducted sensitivity studies with varying levels of VOCs and found the modeled nitrate formation was rather insensitive to the absolute VOC concentrations (Fig. R3). The VOC speciation, especially the fraction of BVOCs, may have an important effect on the nocturnal nitrate formation. At least, the modeling results obtained in the present study should be applicable to the polluted urban atmospheres with little VOC emission in the North China Plain.

We should note that we also didn't consider the impact of the expected changes in the aerosol composition on nitrate formation. We agree with the reviewer that the increasing nitrate may reduce the $N_2O_5$ uptake and to some extent restrict the nocturnal nitrate formation. This issue was not tested in this study.

In the revised manuscript, we have clearly elucidated the limitation of the present modeling analyses, by the following statements.

"It is worth noting that in addition to NOx, $O_3$ and $NH_3$, there are also some other factors that influence the nitrate formation. For example, VOCs are principal $O_3$ precursors, and regulate the abundances of OH and losses of $NO_3$ (and $N_2O_5$). Thus VOCs can affect the daytime $HNO_3$ formation and nocturnal $N_2O_5$ hydrolysis, which in turn affect the nitrate formation. In addition, the increasing nitrate aerosol may reduce the $N_2O_5$ uptake and restrict the nocturnal nitrate formation. These aspects were not quantified in this study. Our modeling analyses were performed with constant VOC level and chemical speciation. Further studies are needed to explore the detailed dependence of nitrate formation to the variety of factors including NOx, $O_3$, $NH_3$, VOCs, aerosol composition and meteorological conditions."

***Typographical and Minor Comments:***

*1:14 – Change 'include the downtown' to 'include locations downtown'*

**Response:** the original sentence has been revised as follows.

"The measurement sites include an urban site in downtown Ji'nan – the capital city of Shandong Province, a rural site downwind of Ji'nan city, and a remote mountain site at Mt. Tai (1534 m a.s.l.)."

*1:18 – Change 'have significantly increased' to 'have statistically significantly increased'*

**Response:** changed.

*1:22 – Change 'at daytime' to 'during the day'. Make this change throughout the entire manuscript (e.g. 2:14, 2:29, 9:14, 12:3, etc.)*

**Response:** this has been changed throughout the entire manuscript.

*1:24 – Reword sentence. Suggest changing to, 'The presence of $NH_3$ contributes to the formation of nitrate aerosol during the day, while decreasing formation at night.*

**Response:** changed as suggested.

*2:2 – Change to 'evidence of a rising trend'*

**Response:** changed.

*2:9 – Remove 'the' before 'climate change'*

**Response:** removed.

*2:14 – Point out that policy mitigation strategies will also depend on understanding aerosol composition and sources.*

**Response:** this sentence has been revised as follows.

"Understanding the chemical composition and sources of atmospheric particles is crucial for quantifying their environmental consequences and formulating science-based mitigation strategies."

*2:15 – Clarify particle phase nitrate vs. gas-phase nitrate radical. i.e. change to 'Particle-phase nitrate ($NO_3^-$) is a principle component…'*

**Response:** changed as suggested.

*2:21-24 – Formation of $NO_3$ and $N_2O_5$ does not only occur at night. Add a sentence clarifying that this process also occurs during the day, but rapid photolysis of $NO_3$ and thermal decomposition of $N_2O_5$ minimize this pathway relative to oxidation of $NO_2$ by the OH radical. Also suggest changing to 'the reaction of $NO_2$ and $O_3$ produce the nitrate radical ($NO_3$), which forms an equilibrium with $N_2O_5$ that can be subsequently taken up onto aerosol to enhance nitrate aerosol.*

**Response:** the original statement has been changed as follows.

"In dark conditions, the reaction of $NO_2$ and $O_3$ produces the nitrate radical ($NO_3$), which forms an equilibrium with $N_2O_5$ that can be subsequently taken up onto particles to enhance nitrate aerosol. The contribution from this pathway is minimized by the rapid photolysis of $NO_3$ and thermal decomposition of $N_2O_5$ during the day."

*2:25 – Change 'nitrogen oxides' to '$NO_x$'*

**Response:** changed.

*2:25-26 – Unclear what the authors mean by 'aqueous transformations of the nitrate radical'. The authors should clarify whether they are referring to $NO_3$ VOC oxidation, which can lead*

*to nitrate containing SOA or direct NO$_3$ uptake onto aerosol.*

**Response:** its means the uptake of NO$_3$ radical onto aerosol and the subsequent aqueous phase reactions with some water-soluble ions and species (see Table S1 for these reactions). For clarity, the original statement has been revised as follows in the revised manuscript.

"There are also some other formation routes of fine nitrate, such as the uptake of NO$_3$ radicals onto aerosols and its subsequent aqueous reactions with some water-soluble species (Hallquist et al., 1999; see also Table S1)."

*3:12 – Change 'depositions' to 'deposition'*

**Response:** changed.

*3:15-17 – Rephrase sentence. Suggest changing to 'In comparison, several recent observational studies have indicated an increasingly important role of aerosol nitrate, which may even dominate summertime haze formation in the NCP'*

**Response:** changed as suggested.

*3:20-23 – Change to 'To the best of our knowledge, there are no previous observational reports of increasing nitrate aerosol over northern China. Long-term measurements are necessary to confirm and quantify this trend, and better understand nitrate formation mechanisms in China.*

**Response:** changed as suggested, thanks.

*3:26 – Change 'mountainous' to 'remote' for consistency*

**Response:** changed.

*3:29 – Change to 'statistically significant'*

**Response:** changed.

*4:4 – Change to 'increasing trend of nitrate aerosol in Northern China,…'*

**Response:** changed.

*4:17 – Change 'last' to 'worst'*

**Response:** changed.

*4:28 – Remove 'due to the closer distance'*

**Response:** removed.

*5:19 – Specify, was particle phase chloride or HCl measured during this study?*

**Response:** both particle phase chloride (Cl$^-$) and gaseous HCl were measured by the MARGA system during this study. This has been specified in the revised manuscript.

*6:16 – Provide the number of chemical reactions in the mechanism to provide the reader with a sense for how explicit daytime VOC degradation is treated.*

**Response:** the model couples the gas-phase RACM2 and aqueous-phase CAPRAM2.4, both of which are connected by a phase transfer module. The RACM2 is a lumped mechanism that consists of 363 chemical reactions to describe the degradation of the variety of VOCs, and the CAPRAM2.4 contains 438 chemical reactions to represent the aqueous reactions of various inorganic and organic compounds. The number of chemical reactions in these mechanisms has been provided in the revised manuscript.

*7:2-3 – Change to 'observed in-situ' and 'available data'*

**Response:** changed.

*7:4-5 – Was a hygroscopic growth factor applied to the aerosol measurements? If so, how was the growth factor curve determined?*

**Response:** The influence of hygroscopic growth was not considered in the previous analysis. In the revised analysis, a hygroscopic growth factor was adopted from Lewis (2008) and Achtert et al. (2009) to take into account the effect of hygroscopic growth on the particle size and surface. All of the modeling analyses were re-performed with the updated particle radius and surface, and the original major conclusions were unchanged. In the revised manuscript, all of the relevant descriptions and results have been updated.

Lewis, E. R.: An examination of Köhler theory resulting in an accurate expression for the equilibrium radius ratio of a hygroscopic aerosol particle valid up to and including relative humidity 100%, J. Geophys. Res., 113, doi: 10.1029/2007jd008590, 2008.

Achtert, P., Birmili, W., Nowak, A., Wehner, B., Wiedensohler, A., Takegawa, N., Kondo, Y., Miyazaki, Y., Hu, M., and Zhu, T.: Hygroscopic growth of tropospheric particle number size distributions over the North China Plain, J. Geophys. Res., 114, 2009.

*7:25-8:5 - Clarify that the reported values are the campaign average ± the standard deviation. Also specify the different years for the Ji'nan results. Since the measurements we not conducted simultaneously, the authors should also discuss expected differences in the reported averages based on the time of year. Lastly, discuss the potential role of atmospheric mixing and transport and how these processes could affect the results at each site.*

**Response:** the original discussion has been revised according to the referee's suggestions. See below for the revised discussion.

"The highest $PM_{2.5}$ levels were recorded at the receptor rural site (Yucheng; with campaign average ± SD of 97.9±53.0 μg m$^{-3}$), followed by the urban (Ji'nan; 68.4±41.7 and 59.3±31.8 μg m$^{-3}$ in 2014 and 2015, respectively) and mountain sites (Mt. Tai; 50.2±31.7 μg m$^{-3}$). Nitrate shows a similar gradient with average concentrations ranging from 6.0±4.6 μg m$^{-3}$ at Mt. Tai to 13.6±10.3 μg m$^{-3}$ at Yucheng."

"It should be noted that these measurements were not conducted simultaneously, and thus difference in the reported data at three study sites can be expected in view of the potential difference in the meteorological conditions which affect atmospheric mixing and transport processes. However, the spatial distributions of emissions, and atmospheric chemical and physical processes are still believed to be the major factor shaping the observed regional

pattern of aerosol pollution."

*8:6 – Clarify what the authors mean by 'different extent of chemical processing'. For example, are the authors referring to $NO_3$ destruction with fresh NO emissions or air transport allowing more processing time?*

**Response:** it means that longer air transport allows more time for chemical processing. This has been clarified in the revised manuscript, see below.

"The air masses sampled at Mt. Tai were more aged and longer air transport allowed more time for chemical processing."

*8:14-17 – Nitrate fractions of 7-14% don't seem to be particularly large and don't 'elucidate the significance of nitrate aerosol in the haze pollution over eastern China'. Perhaps this argument would be more convincing if the authors cited aerosol nitrate fractions from other locations to put these results in context.*

**Response:** the original statements may be misleading for readers who are not very familiar with China. This argument just compares the fraction of nitrate in $PM_{2.5}$ between eastern Chinese cities and a western Chinese city (Xi'an). A recent study reported an average fraction of 7% for $NO_3^-/PM_{2.5}$ in Xi'an, compared to 12-14% in Beijing, Shanghai and Guangzhou, three megacities in eastern China. Our measurements found similar fractions (i.e., 11-14%) of $NO_3^-/PM_{2.5}$ at three sites in the NCP region (also in the east). So we argue that nitrate aerosol may be more important for haze pollution in eastern China compared to western China. For clarity, this argument has been removed from the revised manuscript.

*8:22 – What about the role of ammonium chloride in the calculation of excess $NH_4$? The authors could also look at the molar ratio of total $NH_3$ (g) +$NH_4$ (p) to total $NO_3$ (p) + $HNO_3$ (g) to assess the extent of excess ammonium.*

**Response:** we didn't consider chloride in the calculation of excess $NH_4^+$ given its much lower levels compared to sulfate and nitrate. In the revised analysis, particulate chloride has been taken into account. The updated excess $NH_4^+$ ($18*([NH_4^+]-1.5*[SO_4^{2-}]-[NO_3^-]-[Cl^-])$) were in the range of 0.9-4.3 $\mu g\ m^{-3}$, which were slightly smaller than the estimation without $Cl^-$ (1.4-5.2 $\mu g\ m^{-3}$). We also checked the molar ratios of $([NH_3]+[NH_4^+])/([HNO_3]+[NO_3^-])$, which were in high levels of 9-44 due to high concentrations of $SO_2$, $SO_4^{2-}$ and $NH_3$ (and/or the measurement uncertainties of MARGA for $NH_3$ and $HNO_3$). We have revised the analysis by considering chloride into the excess $NH_4^+$ calculation in the revised manuscript.

"Finally, $NH_4^+$ was generally in excess in $PM_{2.5}$. The average excess $NH_4^+$ (excess $NH_4^+$ =$18*([NH_4^+]-1.5*[SO_4^{2-}]-[NO_3^-]-[Cl^-])$) were calculated in the range of 0.9-4.3 $\mu g\ m^{-3}$ at our three study sites."

*8:27-9:3 – What are the proposed reasons for different diurnal profiles in the urban and rural locations? Is there any available information about the role of mixing nitrate formed aloft down to the surface in the morning?*

**Response:** The main reason might be attributed to the difference in the $NO_2$ patterns in urban and rural locations. The downward mixing of nitrate formed aloft may be an important factor

contributing to the early morning peak of nitrate, although we don't have direct evidence for this from the available observations.

*9:3 – Change to 'The absolute nighttime $NO_3$ levels'*

**Response:** changed.

*9:25 – Was the significance test done at the 95 or 99% level? I.e. is p < 0.05 or 0.01? Make sure this is consistent throughout the text and figures.*

**Response:** the significance test was done at the 99% level with $p<0.01$. The caption in Figure 3 has been revised.

*9:26-27 – Change to 'statistically significant'*

**Response:** changed.

*10:3 – Change to 'Our observations provide direct evidence of a statistically significant increase of summertime nitrate aerosol…'*

**Response:** changed.

*10:11 – Clarify what mitigations strategies have been implemented. This sentence makes it sound as if the entire pollution problem has already been mitigated.*

**Response:** a variety of mitigation strategies have been implemented to cut emissions from industry, transport, biomass burning, road dust, etc., and to optimize the energy structure. As a result, the ambient $PM_{10}$ and $PM_{2.5}$ concentrations have significantly declined in recent several years. However, the entire PM pollution problem has not been thoroughly mitigated. The $PM_{2.5}$ concentrations are still at relatively high levels in some developed regions owning to the abundant secondary components such as nitrate and secondary organic aerosols. The original statements have been revised as follows.

"In recent years, the strict anti-pollution measures implemented by the central government have led to a significant reduction in the primary $PM_{2.5}$ in the NCP, while secondary aerosols such as nitrate are still at high levels and present the major challenge for further mitigation of haze pollution (http://www.cnemc.cn/kqzlzkbgyb2092938.jhml). Nitrate and its precursors should be the next major target for the future control of regional haze pollution in China."

*10:21 – Change to 'deposition'*

**Response:** changed.

*Table S2 – Average is typically abbreviated Avg. not Ave.*

**Response:** corrected.

*11:3 – See comment above, unclear if 'aqueous' $NO_3$ reactions are referring to VOC oxidation and condensation or direct $NO_3$ uptake and reaction.*

**Response:** as discussed above, it refers to the direct $NO_3$ uptake and reaction (see Table S1). The original statement has been revised as follow.

"Hydrolysis of $N_2O_5$ contributed to the remaining (4-6%), and the direct uptake and aqueous-phase reactions of $NO_3$ radicals was negligible."

*11:25-12:11 – Suggest including an example plot in the supplement of the correlation between observed and modeled nitrate aerosol.*

**Response:** we adopt this suggestion and have added the correlation plots between observed and modeled nitrate aerosols for the daytime and nighttime cases in the supplement (see Fig. R1).

---

## Author Comment (AC2) · 14 Jun 2018

**Response to Reviewer 2:**

*The manuscript "Summertime fine particulate nitrate pollution in the North China Plain: Increasing trends, formation mechanisms, and implications for control policy" by Liang Wen and Co-Authors presents the results from measurements conducted in three sites in the North China Plane (urban, rural and remote), in the summertime of 2014 and 2015. Mass and composition of inorganic soluble ions of $PM_{2.5}$ were measured, together with aerosol size distributions, NO, $NO_2$, $O_3$, CO, $SO_2$ concentrations and meteorological parameters. The measurements were compared to previous studies to infer temporal trends of the aerosol nitrate. Additionally, the measurements were compared to the output of the RACM2/CAPRAM2.4 model. The model results were also used to infer the dominant nitrate formation mechanism during the day and at night. Ultimately, the authors performed a sensitivity analysis, modifying the concentrations of precursor gases ($NO_x$ or $NH_3$) in their model to probe which scenario would be the most effective in order to reduce $PM_{2.5}$ pollution in the area.*

*The Referee thinks that the paper addresses relevant scientific questions within the scope of ACP, presenting data of interest to the scientific community. However, 1) the abstract should be rephrased and made clearer; 2) Additional references should be included to give proper credit to related work; 3) some of the methods and assumptions used in the paper should be better outlined and clarified; and 4) some of the figures should be improved for a more straightforward interpretation. The Referee recommends publication in ACP after the comments below are properly addressed.*

**Response:** we thank the referee for the positive comments and helpful suggestions. We have addressed all of the referee's comments in the revised manuscript, as detailed below in the responses to the specific comments. For clarity, the referees' comments are listed below in black italics, while our responses and changes in the manuscript are shown in blue and red, respectively.

*Abstract*

*The Referee thinks that the abstract should be improved. In the current version, a few long sentences and some confusing passages prevent an efficient understanding of the interesting results of the study. In particular:*

*Page 1, Line 14: The Referee suggests breaking the sentence in two parts. One sentence telling about the measurements and one describing the NCP.*

**Response:** this long sentence has been separated into two short ones, as follows.

"The North China Plain (NCP) is one of the most industrialized and polluted regions in China. To obtain a holistic understanding of the nitrate pollution and its formation mechanisms over the NCP region, intensive field observations were conducted at three sites during summertime in 2014-2015."

*Page 1, Line 14-16: the expression "…downtown and downwind Ji'nan…" can be confusing for the Reader that approaches for the first time the description of the measurements sites.*

*Please reword the sentence to make sure that it is clear that those are two distinct sites and that the urban site is downtown Ji'nan and the rural site is downwind of Ji'nan.*

**Response:** the original sentence has been revised as follows.

"The measurement sites include an urban site in downtown Ji'nan – the capital city of Shandong Province, a rural site downwind of Ji'nan city, and a remote mountain site at Mt. Tai (1534 m a.s.l.)."

*Page 1, Line 24-27: The Referee recommends breaking the sentence. One sentence for the day time results and one for the night time results. Additionally, please reword the expression "… plays a slightly negative role…" The word negative is vague and a possible source of confusion for the Reader. Consider using "contributes to a slight decrease in nitrate" or similar.*

**Response:** this sentence has been rephrased as follows.

"The daytime nitrate production in the NCP region is mainly limited by the availability of $NO_2$, and to a lesser extent by $O_3$ and $NH_3$. In comparison, the nighttime formation is controlled by both $NO_2$ and $O_3$. The presence of $NH_3$ contributes to the formation of nitrate aerosol during the day, while slightly decreasing nitrate formation at night."

**Introduction**

*Page 2, Line 21: The authors should consider adding a reference to Song, C. H. and G. R. Carmichael (2001). "Gas-particle partitioning of nitric acid modulated by alkaline aerosol." Journal of Atmospheric Chemistry 40(1): 1-22.*

**Response:** this reference has been added in the revised manuscript.

*Page 2, Line 24: The authors should consider adding a reference to Brown, S. S. and J. Stutz (2012). "Night-time radical observations and chemistry". Chem. Soc. Rev., 41, 6405-6447. doi: 10.1039/c2cs35181a.*

**Response:** added.

*Page 2, Line 25: The authors should consider adding a reference to Dentener, F. J. and P. J. Crutzen (1993). "Reaction of $N_2O_5$ on Tropospheric Aerosols – Impact on the Global Distributions of $NO_x$, $O_3$, and OH." Journal of Geophysical Research- Atmospheres 98(D4): 7149-7163.*

**Response:** added.

**Material and methods**

*Page 7, Line 8-11: The Referee strongly suggests that the Authors indicate the VOC average data used. This is an important information that is omitted in the manuscript and without which it is not possible to reproduce the model results.*

**Response:** the average VOC data used as model inputs in the present study have been provided in the revised supplementary materials (see Table S3).

*Page 7, Line 11-12: The Referee strongly suggests that the Authors indicate the range used for the VOC concentrations in the sensitivity test. Additionally, the statement "...the nitrate formation was insensitive to the input VOC concentrations." should be quantified.*

**Response:** the sensitivity studies were conducted by adjusting the initial VOC concentrations to 0.5 or 1.5 times of the base data, and the model-simulated nitrate increases were compared between the sensitivity tests and base runs. As shown from Figure R1, both sensitivity model runs produced comparable daytime and nocturnal nitrate formation to the base runs (the differences were within 12%). This should be mainly due to the low levels of biogenic VOCs (i.e., isoprene and pinenes) at the study sites, and the reactions of $NO_3$ with BVOCs may only account for a small fraction of the total $N_2O_5$ loss.

[Figure]

**Figure R1.** Sensitivity of the model-simulated (a) daytime and (b) nighttime nitrate formation to the initial VOCs

In the revised manuscript, the original statements have been revised as follows to discuss this aspect, with Figure R1 being added in the supplement.

"The VOC measurements were not made during the present study, and we used the campaign average data previously collected in the same areas during summertime for approximation (Zhu et al., 2016 and 2017). The detailed VOC species and their concentrations as the model input are documented in Table S3. We conducted sensitivity tests with 0.5 or 1.5 times of the initial VOC concentrations, and found that the model simulation was somewhat insensitive to the initial VOC data (the differences between sensitivity tests and base run were within 12%; see Figure S1). This should be mainly due to the low levels of biogenic VOCs in the study area. Given the lack of in-situ VOC measurements, however, the treatment of VOC data presents a major uncertainty in the present modeling analyses."

*Results and discussion*

*In the manuscript, there is no mention of chloride in the aerosol particles. Is it because there was none? The Referee recommends that the Authors add a sentence on the amount of chloride in the particles measured during the study.*

**Response:** we had concurrent chloride data in the present study. It was not mentioned before because we wanted to focus on nitrate in the original analysis. In the revised manuscript, the measured average levels (± standard deviation) of fine particulate $Cl^-$ have been added in

Table 1. The following statement was also added to discuss the amount of chloride measured at three sites in this study.

"Chloride showed comparable levels in urban Ji'nan (1.3±2.1 and 1.3±1.7 µg m$^{-3}$) and rural Yucheng (1.2±1.2 µg m$^{-3}$), with a relatively lower level at Mt. Tai (0.7±0.5 µg m$^{-3}$)."

*Page 8-9, Line 29-2: "…nitrate formation process throughout the nighttime with a $NO_3^-$ increase of 16.9 µg m$^{-3}$…" it is hard to understand where this number comes from. This is because the nighttime is not clearly defined in the manuscript. The Referee suggests to add a definition of night time (maybe using the solar elevation angle) and to add to figure 2 a visual aid (maybe a shaded area) to visually separate nighttime and daytime.*

**Response:** in the revised manuscript, the nighttime period is defined from 19:00 to 7:00 local time. Figure 2 has been improved as suggested to show the nighttime period with shaded areas. However, the 16.9 µg m$^{-3}$ of nitrate increment was calculated from 16:00 to 8:00, which covers the defined night time window. The original statement has been revised as follows in the revised manuscript.

"At Yucheng, the average diurnal profile displays a continuous nitrate formation process throughout the nighttime with a $NO_3^-$ increase of 16.9 µg m$^{-3}$ from 16:00 to 8:00 LT, followed by a sharp decrease during daytime with a trough in the late afternoon (16:00 LT)."

*Page 11, Line 5-11: It is not clear if the RMA slope is from simulated vs observed or vice versa. I guess it is the former case, but it would be advisable to specify if the model over or under predicts the measurements.*

**Response:** yes, it is simulation versus observation. This has been clarified in the revised manuscript.

*Page 11, Line 19-24: I suggest moving this sentence to the next paragraph. The Reader is left hanging at the end of this sentence that, I feel, is a preamble to the first sentence of next paragraph.*

**Response:** we have adopted this suggestion to move these sentences to the next paragraph.

***Figures and Tables***

*The Referee recommends adding an additional table with 3 columns: 1) time of the measurements, 2) location name, and 3) description (urban/rural/remote). This would help the reader navigate the paper more easily.*

**Response:** we have added such a table in the revised supplementary materials (see Table S2).

*Table 1: The Referee thinks it would interesting for the Reader if the Authors would add mean values and standard deviations for $O_3$, $SO_2$, CO, mean diameter and mean number, as the Authors state that those data were available. Additionally, adding the values for the ratio of the sum or the inorganic species divided $PM_{2.5}$ would be a valuable information that would avoid extra work for the reader.*

**Response:** these information have been added to Table 1 in the revised manuscript.

*Figure 2: Please specify if those are averages over all period and add the x-axis label.*

**Response:** these are average diurnal data for the 2014 campaigns. The x-axis label (time of day) has been added.

*Figure 3: Please add x-axis label and standard deviation.*

**Response:** the x-axis label has been added. For the historical data, only average values were taken from the previous literatures, and the standard deviations for some years were not available. Thus standard deviations are not provided in this figure.

*Figure 4 and 5: Please add uncertainty bars to the histograms in the top panel.*

**Response:** added. The uncertainty was expressed here by the standard error of the differences between simulated and observed increase of nitrate aerosol.

*Figure 8 and 9: Please explain in the caption what are the dashed lines.*

**Response:** the dashed lines are only plotted to artificially separate the three zones with distinct sensitivity of nitrate formation to relevant species. This has been explained in figure captions in the revised manuscript.

*Page 2, Line 27 and Page 11, Line 12: I suggest removing "Obviously". It is unnecessary and condescending towards the Reader.*

**Response:** removed.

*Page 3, Line 28: Please specify that in the notation "nitrate/$PM_{2.5}$" and "nitrate/sulfate" the Authors is referring to ratios.*

**Response:** done.

*Page 3, Line 4 and Page 8, Line 10: I suggest removing "relatively". It is unnecessary unless the Authors are able to specify relatively to what.*

**Response:** removed.

---

## Author Comment (AC3) · 14 Jun 2018

**Response to the Editor's Comments:**

*As discussed by the two referees, this paper presents a useful analysis of particulate nitrate pollution in the North China Plain. I concur in their judgment. In addition to the comments of those referees, I suggest that two additional points be discussed:*

**Response:** we thank the editor for handling and evaluating our submission. These comments are very helpful and we have revised the manuscript according to these comments. Below are the responses to the specific comments, with the changes in the manuscript highlighted in red color.

*1) In the Abstract the authors note: "The nitrate/PM$_{2.5}$ and nitrate/sulfate ratios have significantly increased in Ji'nan (2005-2015) and at Mt. Tai (from 2007 to 2014), indicating the worsening situation of regional nitrate pollution." And likewise "This study provides observational evidence of rising trend of nitrate aerosol …" These statements are necessarily correct only if the absolute concentrations of PM$_{2.5}$ and sulfate have remained constant (or increased). If these two species in the denominator of the two ratios have decreased more rapidly than the ratios themselves, then the regional nitrate pollution may be improving in an absolute (but not relative) sense. A short discussion and clarification of this issue should be included.*

**Response:** we explored the trends in the absolute concentrations of PM$_{2.5}$, sulfate and nitrate in Ji'nan and at Mt. Tai. The figures are shown below. Indeed, the ambient levels of PM$_{2.5}$ and sulfate have rapidly decreased in the NCP region over the past decade, largely owing to the stringent control of SO$_2$ emissions and primary particles. In comparison, the absolute concentrations of nitrate in PM$_{2.5}$ showed an increasing trend from 2005 (or 2007) to 2015 (0.29 and 0.39 μg/m$^3$/yr). This confirms the increasing trend of nitrate aerosol pollution in this region. Nevertheless, the available observations since 2011 also showed a decrease in the absolute levels of nitrate aerosol in Ji'nan. This trend may be true given the strict NOx emission control by the central government of China since 2011, but it may be also interfered by the higher aerosol pollution observed during the campaign of 2011 that should be due to the unfavorable meteorological conditions. Anyway, more measurement studies are required to further examine the recent trend of nitrate aerosol since 2011 and assess the impact of the NOx control implemented by the government. The following figures and discussion have been added to clarify this issue in the revised manuscript.

"We also examined the trends in the absolute concentrations of PM$_{2.5}$, nitrate and sulfate in urban Ji'nan and at Mt. Tai (see Fig. S2). As expected, the ambient concentrations of PM$_{2.5}$ (6.3 and 1.4 μg m$^{-3}$ yr$^{-1}$) and SO$_4^{2-}$ (2.1 and 1.2 μg m$^{-3}$ yr$^{-1}$) have rapidly decreased at both locations during the past decade, which should be largely attributed to the stringent control of SO$_2$ emissions and primary particles. In comparison, the absolute concentrations of NO$_3^-$ showed an increasing trend with average rates of change of 0.39 and 0.29 μg m$^{-3}$ yr$^{-1}$. This confirms the increase of nitrate aerosol pollution in the NCP region. Nevertheless, the available observations since 2011 also showed a decrease in the absolute levels of nitrate aerosol in Ji'nan. This trend may be true considering the strict NOx emission control of China since 2011, but it may be also partly interfered by the higher aerosol pollution observed

during the campaign of 2011 with unfavorable meteorological conditions. More measurement efforts are urgently needed to further examine the recent trend of nitrate aerosol after 2011 and evaluate the impact of the NOx emission control implemented by the Chinese government."

[Figure]

**Figure S2.** Long-term trends in the absolute concentrations of (a) $PM_{2.5}$, (b) $NO_3^-$, and (c) $SO_4^{2-}$ in urban Ji'nan and at Mt. Tai in summertime from 2005 to 2015. The fitted lines are derived from the least square linear regression analysis, with the slopes and p values (99% confidence intervals) denoted.

*2) In the Conclusions the authors "recommend that further reduction of anthropogenic emissions of $NO_X$ should be the most efficient pathway for the current control of nitrate aerosol ..." The data discussed in the paper were collected in 2014 and earlier years. Satellite data (e.g., Liu et al., 2017) suggest that $NO_x$ over the North China Plain was increasing during the period covered by these data, but has been decreasing rapidly since 2014. The authors should briefly discuss the likely impact of this $NO_x$ reduction.*

**Response:** indeed, some very recent studies have indicated the decrease in the anthropogenic emissions and ambient abundances of NOx over eastern China in the past five years. It is definitely expected that such reduction of NOx would contribute to a decrease in the fine nitrate aerosol in this region. Nevertheless, this still needs to be further confirmed by the long-term observations in the near future. The following discussion has been added in the revised manuscript.

"Some recent studies have reported the rapid decrease in the NOx abundances over eastern China since 2011 (Liu et al., 2017). It can be expected that such reduction of NOx would help to alleviate the nitrate particulate pollution in China. More observational studies are needed to further examine the trend in the nitrate aerosol and assess the contributions of the strict NOx control of China."

**Reference:**

*Liu, F., Beirle, S., Zhang, Q., van der A, R. J., Zheng, B., Tong, D., & He, K. (2017). $NO_x$ emission trends over Chinese cities estimated from OMI observations during 2005 to 2015. Atmospheric Chemistry and Physics, 17, 9261–9275.*

---

## Author Comment (AC4) · 14 Jun 2018

**Response to Reviewer 3:**

*Fine particulate nitrate pollution has been found to play more and more important role in haze pollution in China. This paper reports measurement results of nitrate and relevant species at three distinctly different sites in the North China Plain, the most polluted region in eastern China, and interprets the main daytime and nighttime formation mechanisms of nitrate and discusses its implications for air pollution measures in this region. This paper gives very important insights into the formation mechanisms of summertime fine particulate nitrate and into the control policy of haze pollution in China. It was very well organized and written and can be accepted for publication in ACP as the following points are addressed.*

**Response:** we thank the reviewer for the positive comments and helpful suggestions. We have addressed all of the following points and revised the original manuscript accordingly. For clarity, the referees' comments are listed below in black italics, while our responses and changes in the manuscript are shown in blue and red, respectively.

*Major points:*

*1) The difference between the Mt. Tai and ground surface sites as well as its implication need to be highlighted. The Mt. Tai site locates around 1465 m a.s.l., which is almost near the top of planetary boundary layer (PBL) in summer. This site is not only a "remote site" in this region, but also can provide more insights into the different chemical mechanisms inside or above the PBL, or in the nocturnal PBL and the residual layer. These issue need to be sharpen in the data analysis or in the discussions.*

**Response:** this point raised by the referee is important and constructive. The Mt. Tai data can indeed provide insights into the chemical conditions in the top boundary layer (daytime) and residual layer (nighttime). Our observations at Mt. Tai demonstrate the serious nitrate aerosol pollution throughout the planetary boundary layer in the NCP region. The nitrate formation mechanisms, including the major formation routes and the sensitivities to NOx, $O_3$ and $NH_3$, were fairly consistent between Mt. Tai and the surface sites. This suggests the regional homogeneity of the in-situ formation of fine nitrate aerosol within the boundary layer in the NCP region. We have added the following discussion about this issue in the revised manuscript.

"It should be noted that the Mt. Tai site is located at around 1465 m a.s.l., which is almost near the top of PBL in summer. Thus the Mt. Tai data can provide insights into the chemical conditions in the top boundary layer at daytime and in the residual layer during the night. Our observations at Mt. Tai demonstrate the serious nitrate aerosol pollution throughout the PBL in the NCP region. Furthermore, the nitrate formation mechanisms, including the major formation routes and sensitivities to NOx, $O_3$ and $NH_3$, were fairly consistent between Mt. Tai and the surface sites. This implies the regional homogeneity in the in-situ formation of fine nitrate aerosol within the PBL over the NCP region."

*2) For the MCM modeling of episodes, the model was run at observational-based mode (OBM). Available measurement data, including nitrate, were used as the model inputs. This method of course could help identify the ongoing chemical processes in the air masses, but it*

*is difficult to trace back to the historical contribution of chemical processes. For example, the observed NH₄NO₃, already existed as initial condition, could be converted into HNO₃ through thermodynamics and further cause an "artificial" mechanism from HNO₃ partitioning. Is that possible to do some sensitivity test by removing or reduction the observed nitrate concentration in the MCM OBM? Otherwise, the authors should mention the weakness or uncertainty of the observational-base modelling when they interpret the modeling results.*

**Response:** we are sorry that the original description of the model setup may be not clear. The RACM-CAPRAM model was only constrained by the hourly measurement data of trace gases and meteorological parameters. The measured aerosol ions data such as nitrate, sulfate and ammonium were only used as initial conditions of the model simulation. The model was initialized with the measured nitrate concentration at the beginning of the episodes, and then simulated the formation of nitrate with constraints of other relevant species. Thus, there should be no artificial mechanism from HNO₃ partitioning with such model setup. We have clarified the detailed model setup by the following statements in the revised manuscript.

"The measured aerosol ions data such as nitrate, sulfate and ammonium were only used as initial conditions of the model simulation. The model was initialized with the measured nitrate concentration at the beginning of the episodes, and then simulated the formation of nitrate with constraints of other relevant species."

***Minor points:***

*1) Please use same scale in Y-axis for the comparison of results from different sites, such as Figure 2, Figure 6 and Figure 7. I understand that the authors would like to highlight some peaks in each panel. However, it is more important to make a comparison between different sites.*

**Response:** these figures have been modified as suggested in the revised manuscript.

*2) About the trends of nitrate/PM₂.₅ and nitrate/sulfate in Figure 3, can we also show the trends of nitrate, NO₂ and O₃ concentration if the data are also available?*

**Response:** we don't have measurement data for NO₂ and O₃ before 2010 in Ji'nan. For Mt. Tai, the measured summertime O₃ levels in 2014 (75±21 ppbv) were comparable to those in 2007 (72±19 ppbv), but the NO₂ measurements were not available in 2007. The nitrate data were available at both sites, and we have plotted the trends of nitrate concentrations in the figure below. This figure has been provided in the revised supplementary materials.

[Figure]

**Figure S2.** Long-term trends in the absolute concentrations of (a) $PM_{2.5}$, (b) $NO_3^-$, and (c) $SO_4^{2-}$ in urban Ji'nan and at Mt. Tai in summertime from 2005 to 2015. The fitted lines are derived from the least square linear regression analysis, with the slopes and p values (99% confidence intervals) denoted.

*3) References of MARGA measurement: Please add some references of measurements based on this instrument, especially those done in the high aerosol loading environment in China.*

**Response:** two references regarding the deployment of MARGA instrument in the polluted environments of China (Wen et al., 2015; Xie et al., 2015) have been cited in the revised manuscript.

Wen, L., Chen, J., Yang, L., Wang, X., Xu, C., Sui, X., Yao, L., Zhu, Y., Zhang, J., Zhu, T., and Wang, W.: Enhanced formation of fine particulate nitrate at a rural site on the North China Plain in summer: The important roles of ammonia and ozone, Atmos. Environ., 101, 294-302, 2015.

Xie, Y., Ding, A., Nie, W., Mao, H., Qi, X., Huang, X., Xu, Z., Kerminen, V.-M., Petäjä, T., Chi, X., Virkkula, A., Boy, M., Xue, L., Guo, J., Sun, J., Yang, X., Kulmala, M., and Fu, C.: Enhanced sulfate formation by nitrogen dioxide: Implications from in situ observations at the SORPES station, J. Geophys. Res., 120, 12679–12694, 10.1002/2015JD02360, 2015.

*4) Page 7, Line 1 and Line 12-14. "Mixing layer height" and "boundary layer height", please use consistent words. In addition, the boundary layer height not only "affects dry deposition", the boundary layer height (or mixing layer height) determines the dispersion capacity of air pollutants emitted from ground surface.*

**Response:** "boundary layer height" has been used in the revised manuscript. We agree that the boundary layer height determines the dispersion capacity of surface air pollutants, but we should note that dispersion was not considered in our box model. The model assumes that the air pollutants are well mixed within the box.

*5) Page 9, line 4-5. The uplifted PBL: the developed PBL or uplifted PBL height.*

**Response:** "the developed PBL" was used as suggested.

---

## Referee Report (RR1)

**Second Review of "Summertime fine particulate nitrate pollution in the North China Plain: Increasing trends, formation mechanisms, and implications for control policy" by Wen, L., et al.**

L. Wen and co-authors have appropriately, thoughtfully, and thoroughly addressed comments and concerns raised in the first review. Below are additional comments on the updated manuscript. Many are editorial in nature, but a few minor details listed below should be addressed prior to acceptance.

**Minor Comments:**
Page 7: line 12 – 7:15 – Additional model and chemical reaction information provided by the authors during revision is very helpful. Please also provide an estimate of the $N_2O_5$ uptake coefficient included in the model and briefly compare to other field-studies. Even though $\gamma(N_2O_5)$ is not explicitly included in the mechanism, I believe this uptake coefficient can be estimated from the $N_2O_5(g) \longleftrightarrow N_2O_5(a)$ rate constant, using: $k\ (s^{-1}) = 0.25*\gamma(N_2O_5)*SA*c$, where SA is the aerosol surface area and c is the mean molecular speed. I suggest adding this information as it will put the magnitude of $N_2O_5$ hydrolysis in this study into context of previous studies.

7:16 – What does the model assume for the deposition of nitrate and $HNO_3$? Dry deposition likely impacts the ground site observations. How do the assumptions pertaining to deposition impact the model results in later sections?

7:29 – What aerosol composition was assumed for the hygroscopic growth calculation?

8:3-8:6 – What method was used for the VOC measurements?

12:19 – How was the early morning period of 06-09:00 LT selected? If the concern is boundary layer expansion and entrainment, this process typically continues past 09:00. If $pNO_3^-$ is mixed down from aloft in the morning (as previous studies have hypothesized), how would this impact the results in this manuscript?

14:7 – What does it mean when the model still predicts nitrate aerosol formation at night when there is no $NH_3$ present in the model (shown in Figure 7)?

15:2 – Cite Roberts 2008 for the current theory on how particle acidity impacts the yield of $ClNO_2$.

> Roberts, J. M., Osthoff, H. D., Brown, S. S., & Ravishankara, A. R. (2008). $N_2O_5$ oxidizes chloride to $Cl_2$ in acidic atmospheric aerosol. *Science, 321*(5892), 1059. https://doi.org/10.1126/science.1158777

16:10 – 16:19 – Thank you to the authors for adding the paragraph on line 17:13. In addition, how sensitive are the results in Figures 7 and 9 to changes in the $N_2O_5$ gas $\rightarrow$ particle conversion rate (i.e. uptake coefficient) and $ClNO_2$ formation rate? In theory, if $N_2O_5$ uptake is inefficient, there will no longer be a linear increase in nitrate with concentrations of $O_3$ and $NO_2$ as shown in Figure 7. Have the authors have considered sensitivity tests to these parameters? In addition to

the added paragraph, the authors should also note that the results in Figure 9 only hold if the sensitivity of nitrate production to $N_2O_5$ uptake does not change under different NOx and $O_3$ conditions. The authors should also clarify that the model simulations are constrained to ground-based observations and the chemistry aloft may show a different sensitivity than in Figures 7 and 9.

Figure 2 – It might be more helpful to use the "Error bars" to plot the standard deviation of each measurement, not the error in the measurement. That way, the variation in the diurnal average profile can be evaluated. I will leave it up to the authors for what they choose to show.

**Editorial Comments:**
1:18 – change to "Using historical observations, the nitrate/$PM_{2.5}$ and…"
2:9 – remove "the" before "Earth's"
2:15 – change to "environmental and health consequences, and…"
2:26 – Move "during the day" to after "minimized"
3:4-3:10 – Switch the order of the sentences starting on line 3:4, "Field measurements…" and on line 3:7, "The contribution…".
3:6 – Add McDuffie et al., 2018 and Tham et al., 2018 to the Brown and Stutz reference, since both papers provide overviews of the current state of agreement between field-derived uptake coeffects and laboratory-based parameterizations.

McDuffie, E. E., Fibiger, D. L., Dubé, W. P., Lopez-Hilfiker, F., Lee, B. H., Thornton, J. A., et al. (2018). Heterogeneous $N_2O_5$ uptake during winter: Aircraft measurements during the 2015 WINTER campaign and critical evaluation of current parameterizations. *Journal of Geophysical Research: Atmospheres*. https://doi.org/10.1002/2018JD028336

Tham, Y. J., Wang, Z., Li, Q., Wang, W., Wang, X., Lu, K., et al. (2018). Heterogeneous $N_2O_5$ uptake coefficient and production yield of $ClNO_2$ in polluted northern China: Roles of aerosol water content and chemical composition. *Atmospheric Chemistry and Physics Discussions, 2018*, 1-27. https://doi.org/10.5194/acp-2018-313

3:7-3:9 – After the Baasandorj reference, add ", but will be dependent on the rate of $NO_3$ formation and reaction, and the $N_2O_5$ uptake coefficient ($\gamma(N_2O_5)$) and formation yield of $ClNO_2$."
3:9 – 3:10 – Add the following references to the Baasandorj reference, which all discuss the vertical transport of nitrate aerosol:

Brown, S. G., Hyslop, N. P., Roberts, P. T., McCarthy, M. C., & Lurmann, F. W. (2006). Wintertime Vertical Variations in Particulate Matter (PM) and Precursor Concentrations in the San Joaquin Valley during the California Regional Coarse PM/Fine PM Air Quality Study. *Journal of the Air & Waste Management Association, 56*(9), 1267-1277. https://doi.org/10.1080/10473289.2006.10464583

Prabhakar, G., Parworth, C. L., Zhang, X., Kim, H., Young, D. E., Beyersdorf, A. J., et al. (2017). Observational assessment of the role of nocturnal residual-layer chemistry in

determining daytime surface particulate nitrate concentrations. *Atmospheric Chemistry and Physics, 17*(23), 14747-14770. https://doi.org/10.5194/acp-17-14747-2017

Pusede, S. E., Duffey, K. C., Shusterman, A. A., Saleh, A., Laughner, J. L., Wooldridge, P. J., et al. (2016). On the effectiveness of nitrogen oxide reductions as a control over ammonium nitrate aerosol. *Atmospheric Chemistry and Physics, 16*(4), 2575-2596. https://doi.org/10.5194/acp-16-2575-2016

Watson, J. G., & Chow, J. C. (2002). A wintertime PM$_{2.5}$ episode at the Fresno, CA, supersite. *Atmospheric Environment, 36*(3), 465-475. https://doi.org/https://doi.org/10.1016/S1352-2310(01)00309-0

3:19 – Change to "about a 75% reduction"
4:15 – insert "the" before "North China Plain"
5:15 – Insert "the" before "mountain peak"
5:17 – Change to "descriptions"
6:11 – Change to "quantified *in-situ*"
6:20 – Remove "well qualified and"
6:27 – Remove "the" before "gas-and aqueous…"
7:12 – Add Tham et al., 2018 and McDuffie et al., 2018 from above to the Chang 2011 reference. These studies provide information on the up-to-date status of field-parameterization differences.
7:15 – Change to "utilized previously to simulate…"
8:9 – Clarify what "differences" you are referring to
8:14 – Change to "Simulations were conducted…"
8:16 – Change to "major aerosol formation…"
9:14 – Change to "differences"
9:15 – Change to "differences"
10:1 – Add "power" before "plant"
10:3 – Change "were" to "was"
10:14 – Change to" "thermal decomposition of aerosol"
11:7 – Change to "derived at Mt. Tai from data collected in 2007 and 2014, affirming…" This clarifies that this trend is based on two years of data.
11:18 – Add at the end of the sentence, "at Ji'nan and Mt. Tai, respectively".
11:21 and 11:25 – Subscript NO$_x$
12:5 – Change "a more and more" to "an increasingly"
13:10 – Change "over" to "out"
16:4 – Remove "to be"
17:7 – Change "that" to "the"
17:17 – Add appropriate references for that statement that increasing aerosol nitrate may reduce the N$_2$O$_5$ uptake coefficient.
15:18 – Change "series" to "serious"
Table S1 – Are the units in cm$^3$ molecules$^{-1}$ s$^{-1}$? If so, change "mol" to "molec.". If not, disregard this comment.
Figures 8 & 9 – label the three sensitivity regimes on the contour plots

---

## Author Response (AR2)

**Response to Reviewer's Comments**

***Second Review of "Summertime fine particulate nitrate pollution in the North China Plain: Increasing trends, formation mechanisms, and implications for control policy" by Wen L., et al.***

*L. Wen and co-authors have appropriately, thoughtfully, and thoroughly addressed comments and concerns raised in the first review. Below are additional comments on the updated manuscript. Many are editorial in nature, but a few minor details listed below should be addressed prior to acceptance.*

**Response:** we thank the reviewer for the further evaluation and helpful comments on our revised manuscript. Below we address all of these comments point by point, and the manuscript has been further revised accordingly. Again, the referees' comments are listed in black italics, while our responses and changes in the manuscript are shown in blue and red, for clarity.

**Minor Comments:**
*Page 7: line 12 – 7:15 – Additional model and chemical reaction information provided by the authors during revision is very helpful. Please also provide an estimate of the $N_2O_5$ uptake coefficient included in the model and briefly compare to other field-studies. Even though $\gamma(N_2O_5)$ is not explicitly included in the mechanism, I believe this uptake coefficient can be estimated from the $N_2O_5(g) \leftrightarrow N_2O_5(a)$ rate constant, using: $k\ (s^{-1}) = 0.25*\gamma(N_2O_5)*SA*c$, where SA is the aerosol surface area and c is the mean molecular speed. I suggest adding this information as it will put the magnitude of $N_2O_5$ hydrolysis in this study into context of previous studies.*

**Response:** we thank the reviewer for this very good comment. According to the reviewer's suggestion, we have estimated the $\gamma(N_2O_5)$ values from the $N_2O_5(g) \leftrightarrow N_2O_5(a)$ rate constant and the measured aerosol surface area, and the average $\gamma(N_2O_5)$ ($\pm$SD) for our selected cases was 0.018$\pm$0.00006. Such $\gamma(N_2O_5)$ values are well within the reported ranges of $\gamma(N_2O_5)$ derived from field observations in other locations worldwide (e.g., 0.001-0.1; Tham et al., 2018 and references therein), and are comparable to or slightly lower than those derived at several sites in northern China, i.e., Mt. Tai (0.021-0.103), Wangdu (0.006-0.034), Beijing (0.012-0.055), and Ji'nan (0.042-0.092). The following information has been added in the revised manuscript.

"We estimated the $\gamma(N_2O_5)$ from the reaction rate for the $N_2O_5$ gas-to-particle partitioning and the measured aerosol surface area concentrations, and derived an average $\gamma(N_2O_5)$ ($\pm$SD) of 0.018$\pm$0.00006 for our selected cases. Such levels are well within the reported range of $\gamma(N_2O_5)$ derived from the field observations in other locations worldwide (e.g., 0.001-0.1), including several polluted areas in northern China (Tham et al., 2018; and references therein)."

*7:16 – What does the model assume for the deposition of nitrate and HNO₃? Dry deposition likely impacts the ground site observations. How do the assumptions pertaining to deposition impact the model results in later sections?*

**Response:** the model doesn't consider the deposition of nitrate aerosol, but considers the dry deposition of $HNO_3$. The deposition velocity of $HNO_3$ was set as 2 cm $s^{-1}$, and the boundary layer height was set to vary from 200 m to ~1300 m for our cases in the model. We compared the loss rates of $HNO_3$ from dry deposition and from the $HNO_3$ gas-to-particle partitioning, and found dry deposition only presented a very minor fraction of the total $HNO_3$ sink (<1%). Therefore, dry deposition should not affect the subsequent modelling results in this study. In the revised manuscript, the following statements have been added to clarify this issue.

"The dry deposition velocity of $HNO_3$ was set as 2 cm $s^{-1}$ in the model. With such configuration, dry deposition only presents a minor fraction of the daytime $HNO_3$ sink (<1%), compared to the $HNO_3$ gas-to-particle partitioning."

*7:29 – What aerosol composition was assumed for the hygroscopic growth calculation?*

**Response:** we just took the parameterization from literature, which was derived from field observations at a rural site of Beijing. Thus it is based on the aerosol composition measured at a rural site in the North China Plain. The original statement has been revised as follows to elaborate this in the revised manuscript.

"A hygroscopic growth factor obtained from the NCP region by Achtert et al. (2009) was adopted to take into account the effect of hygroscopic growth on particle size and surface."

Achtert, P., Birmili, W., Nowak, A., Wehner, B., Wiedensohler, A., Takegawa, N., Kondo, Y., Miyazaki, Y., Hu, M., and Zhu, T.: Hygroscopic growth of tropospheric particle number size distributions over the North China Plain, J. Geophys. Res., 114, doi: 10.1029/2008jd010921, 2009.

*8:3-8:6 – What method was used for the VOC measurements?*

**Response:** the VOC measurements at Mt Tai and Yucheng were made off-line based on the canister sampling coupled with analysis by GC+FID/MS. At Ji'nan, the VOC measurements were conducted by an online GC+FID analyzer. We have added this information in Table S3 in the revised manuscript.

12:19 – How was the early morning period of 06-09:00 LT selected? If the concern is boundary layer expansion and entrainment, this process typically continues past 09:00. If $p(NO_3^-)$ is mixed down from aloft in the morning (as previous studies have hypothesized), how would this impact the results in this manuscript?

**Response:** to be honest, the early morning period of 06:00-09:00 LT was arbitrarily selected

in the analysis. However, we think this should not affect the analysis results in this manuscript, because the observed increase of nitrate aerosol during these selected cases were only used to compare with the model-simulated p($NO_3^-$). All of the subsequent analyses were based on the modelling results (including only chemistry). Furthermore, the observed nitrate increments during these selected periods show quite good correlations with the model-simulated p($NO_3^-$). In the revised manuscript, the following statement has been modified to clarify this.

"4) the data in the early morning period (i.e., 06:00-09:00 LT) were excluded from analyses to roughly eliminate the potential influence from downward mixing of air aloft to the surface sites."

*14:7 – What does it mean when the model still predicts nitrate aerosol formation at night when there is no $NH_3$ present in the model (shown in Figure 7)?*

**Response:** it means that the nitrate formation from the hydrolysis of $N_2O_5$ is not sensitive to the availability of $NH_3$. This has been discussed in the manuscript. Anyway, the model was actually initialized with an amount of $NH_4^+$ in the aerosol phase.

*15:2 – Cite Roberts 2008 for the current theory on how particle acidity impacts the yield of $ClNO_2$.*

*Roberts, J. M., Osthoff, H. D., Brown, S. S., & Ravishankara, A. R. (2008). $N_2O_5$ oxidizes chloride to $Cl_2$ in acidic atmospheric aerosol. Science, 321(5892), 1059. https://doi.org/10.1126/science.1158777*

**Response:** this reference has been cited in the revised manuscript.

*16:10 – 16:19 – Thank you to the authors for adding the paragraph on line 17:13. In addition, how sensitive are the results in Figures 7 and 9 to changes in the $N_2O_5$ gas → particle conversion rate (i.e. uptake coefficient) and $ClNO_2$ formation rate? In theory, if $N_2O_5$ uptake is inefficient, there will no longer be a linear increase in nitrate with concentrations of $O_3$ and $NO_2$ as shown in Figure 7. Have the authors have considered sensitivity tests to these parameters? In addition to the added paragraph, the authors should also note that the results in Figure 9 only hold if the sensitivity of nitrate production to $N_2O_5$ uptake does not change under different NOx and $O_3$ conditions. The authors should also clarify that the model simulations are constrained to ground-based observations and the chemistry aloft may show a different sensitivity than in Figures 7 and 9.*

**Response:** we have not considered the sensitivity tests to these parameters. As shown in the response to the reviewer's first comment, existing field studies in the North China Plain have found fast heterogeneous uptake of $N_2O_5$ onto particles, with field-derived $\gamma(N_2O_5)$ values of 0.042-0.092 in Ji'nan, 0.021-0.103 at Mt. Tai, 0.006-0.034 at Wangdu, and 0.012-0.055 in Beijing. The estimated average $\gamma(N_2O_5)$ value used in our model was 0.018, which was even smaller than these values obtained from field observations. The $ClNO_2$ formation depends on

the measured levels of Cl[-] and the explicit aqueous-phase reactions of $NO_2^+$ with Cl[-]. Thus the $N_2O_5$ uptake should be efficient in the NCP region, and our representation in the model may be a lower estimation of the $N_2O_5$ uptake process. Besides, the clarifications suggested by the reviewer have been added in the revised manuscript. See below.

"The results in Figure 9 only hold if the sensitivity of nitrate production to $N_2O_5$ uptake does not change under different NOx and $O_3$ conditions. Furthermore, the model simulations are constrained to ground-based observations and the chemistry aloft may show a different sensitivity than in Figures 7 and 9. These aspects were not quantified in this study. Further studies are needed to explore the detailed dependence of nitrate formation to the variety of factors including NOx, $O_3$, $NH_3$, VOCs, aerosol composition, and meteorological conditions."

*Figure 2 – It might be more helpful to use the "Error bars" to plot the standard deviation of each measurement, not the error in the measurement. That way, the variation in the diurnal average profile can be evaluated. I will leave it up to the authors for what they choose to show.*

**Response:** Figure 2 has been modified as suggested, with standard deviations being plotted.

***Editorial Comments:***
*1:18 – change to "Using historical observations, the nitrate/$PM_{2.5}$ and…"*

**Response:** changed.

*2:9 – remove "the" before "Earth's"*

**Response:** removed.

*2:15 – change to "environmental and health consequences, and…"*

**Response:** changed.

*2:26 – Move "during the day" to after "minimized"*

**Response:** done.

*3:4-3:10 – Switch the order of the sentences starting on line 3:4, "Field measurements…" and on line 3:7, "The contribution…"*

**Response:** done.

*3:6 – Add McDuffie et al., 2018 and Tham et al., 2018 to the Brown and Stutz reference, since both papers provide overviews of the current state of agreement between field-derived uptake coefficients and laboratory-based parameterizations.*

*McDuffie, E. E., Fibiger, D. L., Dubé W. P., Lopez-Hilfiker, F., Lee, B. H., Thornton, J. A., et al. (2018). Heterogeneous $N_2O_5$ uptake during winter: Aircraft measurements during the 2015 WINTER campaign and critical evaluation of current parameterizations. Journal of Geophysical Research: Atmospheres. https://doi.org/10.1002/2018JD028336*

*Tham, Y. J., Wang, Z., Li, Q., Wang, W., Wang, X., Lu, K., et al. (2018). Heterogeneous $N_2O_5$ uptake coefficient and production yield of $ClNO_2$ in polluted northern China: Roles of aerosol water content and chemical composition. Atmospheric Chemistry and Physics Discussions, 2018, 1-27. https://doi.org/10.5194/acp-2018-313*

**Response:** these latest references have been added.

*3:7-3:9 – After the Baasandorj reference, add ", but will be dependent on the rate of $NO_3$ formation and reaction, and the $N_2O_5$ uptake coefficient ($\gamma(N_2O_5)$) and formation yield of $ClNO_2$."*

**Response:** this statement has been added.

*3:9 – 3:10 – Add the following references to the Baasandorj reference, which all discuss the vertical transport of nitrate aerosol:*

*Brown, S. G., Hyslop, N. P., Roberts, P. T., McCarthy, M. C., & Lurmann, F. W. (2006). Wintertime Vertical Variations in Particulate Matter (PM) and Precursor Concentrations in the San Joaquin Valley during the California Regional Coarse PM/Fine PM Air Quality Study. Journal of the Air & Waste Management Association, 56(9), 1267-1277. https://doi.org/10.1080/10473289.2006.10464583*

*Prabhakar, G., Parworth, C. L., Zhang, X., Kim, H., Young, D. E., Beyersdorf, A. J., et al. (2017). Observational assessment of the role of nocturnal residual-layer chemistry in determining daytime surface particulate nitrate concentrations. Atmospheric Chemistry and Physics, 17(23), 14747-14770. https://doi.org/10.5194/acp-17-14747-2017*

*Pusede, S. E., Duffey, K. C., Shusterman, A. A., Saleh, A., Laughner, J. L., Wooldridge, P. J., et al. (2016). On the effectiveness of nitrogen oxide reductions as a control over ammonium nitrate aerosol. Atmospheric Chemistry and Physics, 16(4), 2575-2596. https://doi.org/10.5194/acp-16-2575-2016*

*Watson, J. G., & Chow, J. C. (2002). A wintertime PM2.5 episode at the Fresno, CA, supersite. Atmospheric Environment, 36(3), 465-475. https://doi.org/https://doi.org/10.1016/S1352-2310(01)00309-0*

**Response:** all of these references have been added in the revised manuscript.

*3:19 – Change to "about a 75% reduction"*

**Response:** changed.

*4:15 – insert "the" before "North China Plain"*

**Response:** done.

*5:15 – Insert "the" before "mountain peak"*

**Response:** done.

*5:17 – Change to "descriptions"*

**Response:** changed.

*6:11 – Change to "quantified in-situ"*

**Response:** changed.

*6:20 – Remove "well qualified and"*

**Response:** removed.

*6:27 – Remove "the" before "gas-and aqueous…"*

**Response:** removed.

*7:12 – Add Tham et al., 2018 and McDuffie et al., 2018 from above to the Chang 2011 reference. These studies provide information on the up-to-date status of field-parameterization differences.*

**Response:** added.

*7:15 – Change to "utilized previously to simulate…"*

**Response:** changed.

*8:9 – Clarify what "differences" you are referring to*

**Response:** it refers to the difference in the model-simulated nitrate increment (formation) between base and sensitivity simulations. This has been clarified in the revised manuscript.

*8:14 – Change to "Simulations were conducted…"*

**Response:** changed.

*8:16 – Change to "major aerosol formation…"*

**Response:** changed.

*9:14 – Change to "differences"*

**Response:** changed.

*9:15 – Change to "differences"*

**Response:** changed.

*10:1 – Add "power" before "plant"*

**Response:** added.

*10:3 – Change "were" to "was"*

**Response:** changed.

*10:14 – Change to" "thermal decomposition of aerosol"*

**Response:** changed.

*11:7 – Change to "derived at Mt. Tai from data collected in 2007 and 2014, affirming…"*
*This clarifies that this trend is based on two years of data.*

**Response:** changed.

*11:18 – Add at the end of the sentence, "at Ji'nan and Mt. Tai, respectively".*

**Response:** added.

*11:21 and 11:25 – Subscript NOx*

**Response:** done.

*12:5 – Change "a more and more" to "an increasingly"*

**Response:** changed.

*13:10 – Change "over" to "out"*

**Response:** changed.

*16:4 – Remove "to be"*

**Response:** removed.

*17:7 – Change "that" to "the"*

**Response:** changed.

*17:17 – Add appropriate references for that statement that increasing aerosol nitrate may reduce the $N_2O_5$ uptake coefficient.*

**Response:** the reference of Chang et al. 2011 has been added in the revised manuscript.

Chang, W. L., Bhave, P. V., Brown, S. S., Riemer, N., Stutz, J., and Dabdub, D.: Heterogeneous Atmospheric Chemistry, Ambient Measurements, and Model Calculations of $N_2O_5$: A Review, Aerosol Sci. Tech, 45, 665-695, 10.1080/02786826.2010.551672, 2011.

*18:18 – Change "series" to "serious"*

**Response:** changed.

*Table S1 – Are the units in $cm^3$ $molecules^{-1}$ $s^{-1}$? If so, change "mol" to "molec.". If not, disregard this comment.*

**Response:** the units are $cm^3$ $mole^{-1}$ $s^{-1}$ for the aqueous phase reactions, not the $cm^3$ $molecule^{s-1}$ $s^{-1}$.

*Figures 8 & 9 – label the three sensitivity regimes on the contour plots.*

**Response:** done.

---

## Author Response (AR3)

**Response to the Editor's Comments:**

*This paper has been exemplary in the publishing process. A good paper was submitted, reviewers provided helpful comments, which the authors fully considered, and a very good paper has resulted. The paper is ready for publication after one very minor question is addressed. The revision has added the phrase: "...derived an average γ(N₂O₅) (±SD) of 0.018±0.00006 for our selected cases." Is the standard deviation really more than 2 orders of magnitude smaller than the average?*

**Response:** we thank the editor so much for your great efforts for handling and evaluating our manuscript. The review and publishing process is very helpful for improving our research. For the comment on $\gamma(N_2O_5)$, we checked again the calculation results and found that the derived $\gamma(N_2O_5)$ values are quite uniform, i.e., in the range of 0.0178–0.0179. So the standard deviation is very small. We have further revised the original phrase as follows.

"We estimated the $\gamma(N_2O_5)$ from the reaction rate for the $N_2O_5$ gas-to-particle partitioning and the measured aerosol surface area concentrations, and derived a $\gamma(N_2O_5)$ value of 0.018 for our selected cases. Such level is well within the reported range of $\gamma(N_2O_5)$ derived from the field observations in other locations worldwide (e.g., 0.001-0.1), including several polluted areas in northern China (Tham et al., 2018; and references therein)."